# 3D deep geothermal reservoir imaging with wireline distributed acoustic sensing in two boreholes.

Evgeniia Martuganova[1,2], Manfred Stiller[1], Ben Norden[1], Jan Henninges[1,3], and Charlotte M. Krawczyk[1,2]

1 Helmholtz Centre Potsdam, GFZ German Research Centre for Geosciences,14473 Potsdam, Germany

2 Department of Applied Geophysics, Technische Universität Berlin,10587 Berlin, Germany

3 Federal Office for the Safety of Nuclear Waste Management (BASE), 10623 Berlin, Germany

**Correspondence:** Evgeniia Martuganova (e.martuganova@outlook.com)

**Abstract.** Geothermal exploration will help moving towards a low-carbon economy and provide a basis for green and sustainable growth. The development of new practical, reliable methods for geophysical characterisation of a reservoir has the potential to facilitate a broader application of deep geothermal energy. At the Groß Schönebeck in-situ laboratory, a unique vertical seismic profiling (VSP) dataset was recorded in two 4.3 km deep geothermal boreholes using fibre optic cables in early 2017. The experiment set-up consisted of 61 vibrator points organised in a spiral pattern around the well site to ensure a proper offset and azimuth distribution in the target reservoir section. Data were processed using a standard workflow for VSP. As a result, a detailed 3-dimensional 0.75 x 1 x 4.5 km size image around the existing boreholes was created using the Kirchhoff migration algorithm with restricted aperture. The imaging resolved small-scale features in the reservoir essential for the future exploration of the geothermal research site. Borehole data with vertical resolution up to 16 m revealed the existing depth variations of the Elbe reservoir sandstone horizon at 4.08 - 4.10 km depth and indications of an unconformity in the area where we expect volcanic rocks. In addition, in the borehole data a complex interlaying with numerous pinch outs in the Upper Rotliegend reservoir section (3.8 to 4 km depth) was discovered. Thereby, we demonstrate that wireline fibre optic data can significantly contribute to exploration by providing an efficient and reliable method for deep geothermal reservoir imaging.

## 1 Introduction

The EU aims to expand geothermal energy use and reach 2570 TWh by 2050 (The European Commission, 2021). Nevertheless, in Europe, the amount of easily accessible hydrothermal resources are limited. Consequently, the primary growth in geothermal power production is expected to come from projects associated with developing enhanced geothermal systems (EGS) (Carrara et al., 2020). The development of geothermal plants which exploit EGS reservoirs is associated with increased developing costs, high risks associated with the drilling of deep wells, and the possible requirement of well stimulation (Carrara et al., 2020). Due to these reasons, EGS plants are quite rare within the EU (IEA Geothermal, 2020). To overcome the high upfront investment costs barrier and to support the growing demand for the exploration of deep geothermal reservoirs (3.5-4 km), it is essential to develop modern, reliable technological solutions to reduce costs and risks related to the deep geothermal wells

drilling and EGS plants installations. According to the geothermal energy technology development report 2020 (Carrara et al., 2020), geothermal exploration could be optimised by utilising new methods, i.e., applying fibre optics cables to measure strain.

Measurements with a fibre-optic cable installed in diverse environments are widely applied for seismic data acquisition for versatile research topics such as glacial studies (Booth et al., 2020; Brisbourne et al., 2021; Hudson et al., 2021), volcanology (Currenti et al., 2021; Klaasen et al., 2021; Jousset et al., 2022), underwater seismology (Spica et al., 2020a; Lior et al., 2021) and urban seismology (Dou et al., 2017; Spica et al., 2020b; Yuan et al., 2020). Perhaps, one of the most well-studied distributed acoustic sensing (DAS) applications is cable deployment for data acquisition in boreholes. Measurements with

a fibre-optic cable installed along the casing or behind the tubing are widely and successfully applied for borehole seismic data acquisition. 3D vertical seismic profiling (VSP) imaging results with permanent cable installation include applications for oil and gas exploration (Jiang et al., 2016; Shultz and Simmons, 2019; Zhan and Nahm, 2020), the monitoring of $CO_2$ reservoirs (Humphries et al., 2015; Götz et al., 2018; Correa et al., 2019; Wilson et al., 2021), and for mineral exploration and mining (Bellefleur et al., 2020). Nevertheless, in a vast number of already drilled and cased boreholes, only measurements with

a wireline logging cable are feasible. Surveys with retrievable fibre optic cable, such as the dataset collected for petroleum exploration in China, near Tangshan, Hebei (Yu et al., 2016), are still quite unique and rarely found in the peer-reviewed literature.

    Geothermal exploration sites pose additional challenges in terms of the requirements for the instrumentation. Fibre optic cables have increased durability and are less subjected to corrosion compared to conventional sensors (Reinsch et al., 2015).

Therefore seismic data acquisition using DAS becomes feasible for harsh conditions (elevated pressure, salinity, temperature, high acidity) for extended time periods. One of the essential aspects of the EGS project development is microseismic monitoring during hydraulic fracturing. Lellouch et al. (2021) demonstrated that a vertical downhole DAS array could be successfully used in the subsurface with elevated temperatures up to 175°C to detect low-magnitude earthquakes at a range of up to 10 km from the borehole location in the Frontier Observatory for Research in Geothermal Energy (FORGE) site in Utah. Fibre optic cable

buried at a depth of 0.5 m at Brady geothermal field allowed to record a substantial amount of earthquakes and provided information on the evolution of the seismicity during geothermal plant operation (Li and Zhan, 2018). Using optical cables allows surveys with dense spacing, which are very expensive and often cost prohibited for geothermal applications in case of data acquisition with conventional methods. A detailed image of the subsurface can be created using various seismic imaging techniques (Krawczyk, 2021). Nevertheless, DAS VSP surveys for geothermal are still rarely acquired (Miller et al., 2018;

Trainor-Guitton et al., 2018). Miller et al. (2018) demonstrate limited results in the form of unmigrated seismograms with a maximum depth of 297 m. In the case of the 3D imaging results presented from the Brady geothermal field, the major difference is in depth, which is limited to 600 m in this case study (Trainor-Guitton et al., 2018).

    Although there are a few examples of geothermal exploration applications, only one shallow geothermal VSP has thus far been reported in the literature with 3D imaging results. Constantinou et al. (2016) showed a test wireline DAS-VSP dataset

with a maximum surveyed depth of 2580 m MD acquired at Rittershoffen geothermal site, however, there has not been any 3D imaging results reported from a deep geothermal well using wireline DAS VSP to the best of the author's knowledge. Moreover, only a very limited number of publications on wireline DAS applications with active seismic sources can be found

in the literature. Therefore, further thorough research on the evaluation of the wireline DAS data in geothermal applications is needed.

This paper presents results from one of the first applications of DAS VSP for deep geothermal exploration at the Groß Schönebeck geothermal research site down to 4 km depth. First, we evaluate the acquired data and demonstrate cable installation's influence on the data quality. Then, after a brief explanation of the processing flow, the 3D DAS VSP imaging results at Gross Schoenebeck will be presented, followed by geological interpretation. In conclusion, we will focus on the deliverables of the 3D DAS VSP and how this can contribute to the characterisation of the reservoir and geothermal exploration.

## 2   The Groß Schönebeck site

The in-situ laboratory Groß Schönebeck is located in the Northern German Basin, one of Germany's main regions with deep hydrothermal resources. The joint research project RissDom-A (RissDominierte Erschließung in German: fracture-dominated exploitation) aims to gain expertise in sustainable energy production from low permeability geothermal reservoirs by developing Enhanced Geothermal Systems (EGS). The Buntsandstein sandstone formation and volcanic rocks of Lower Permian
(Rotliegend) (Figure 1a) age are of interest for the direct use for the geothermal energy production (Blöcher et al., 2016). A successful geothermal exploration case study can lead to broader geothermal energy usage in the regions without hydrothermal potential. Moreover, the geological setting at the experiment site is typical for a broad part of Northern Europe. Therefore, the acquired knowledge from this case study can be applied to geothermal exploration programs in other areas with similar geological conditions.

To deepen the understanding of the geological structures interpreted on sparse vintage 2D seismic lines and locate possible faults within the area of interest, a high-resolution 3D reflection seismics acquisition campaign was carried out in February/March 2017 (Stiller et al., 2018; Krawczyk et al., 2019). The 3D surface seismic survey was designed to cover an area of 8 km by 8 km, focusing on target reservoir depth from 4 to 4.3 km. Since the studied geothermal reservoir zone is located at a depth greater than 4 km and overlaid by Zechstein salt, it is a challenging target for exploration with conventional seismic
methods.

### 2.1   The distributed acoustic sensing vertical seismic profiling survey

In contrast to surface seismics, VSP has the advantage of shorter reflection travel paths. Therefore, the amplitudes are theoretically less subjected to attenuation. As a result, this provides a better signal-to-noise ratio and broader frequency content. Thus, to improve the resolution of seismic data, which is limited due to the thick salt layer (more than 1 km), and to provide
detailed imaging around the existing boreholes, an extensive VSP experiment using wireline DAS technology was conducted prior the main surface seismic experiment in February 2017. Two wireline hybrid cables (electrical and optical; from Schlumberger (NOVA-F) and GFZ (Rochester)) (Henninges et al., 2011) were temporarily deployed and hanging freely with 1 m slack applied inside the casing of two deep water-filled boreholes. Nearly vertical well E GrSk 3/90 (maximum inclination 7.2°), which was formerly used for oil and gas exploration, and Gt GrSk 4/05 (maximum inclination 49°) form a geothermal doublet

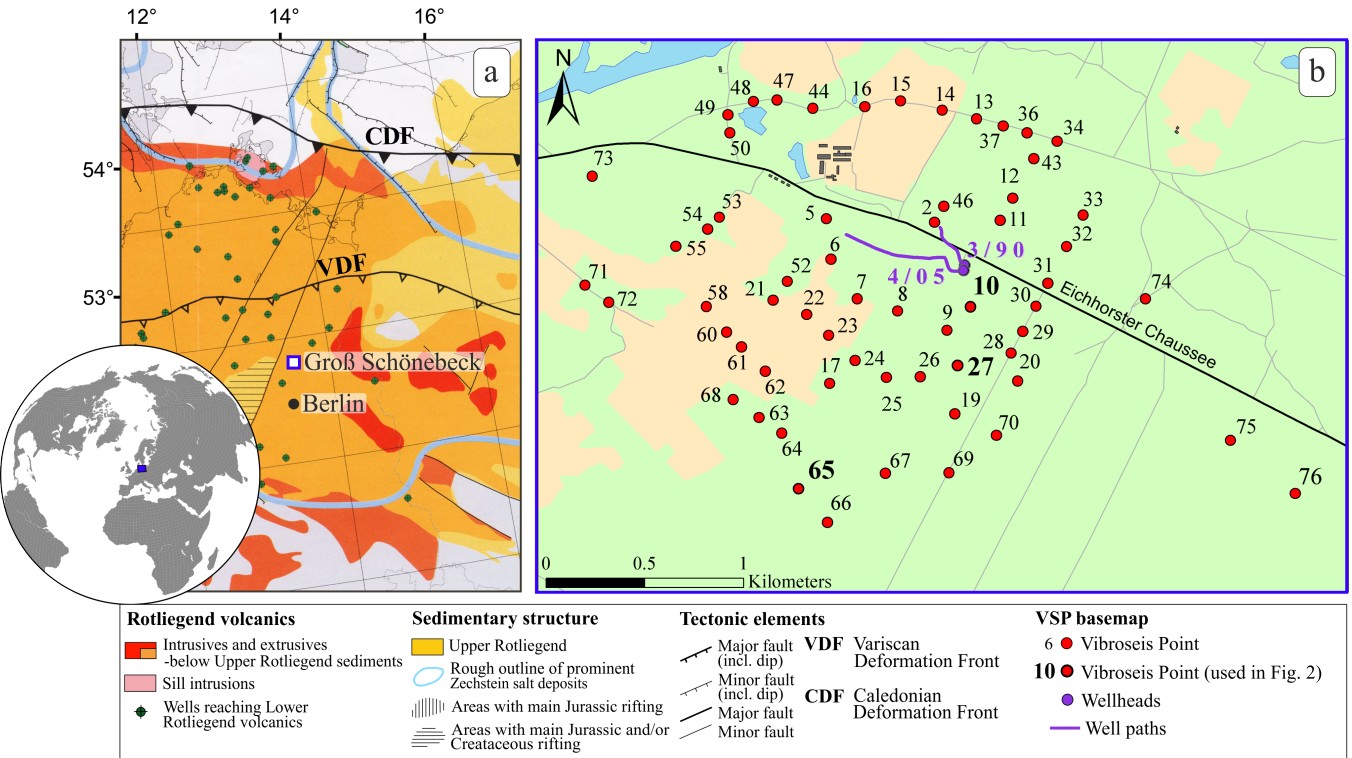

**Figure 1.** Location of the Groß Schönebeck geothermal site. (a) Rotliegend volcanics and sedimentary distribution in the Southern Permian Basin (compilation from Krawczyk et al. (2019)). (b) Base map of the Groß Schönebeck geothermal site with two boreholes (violet circles) in the centre and 61 VSP source points (red dots) arranged in a spiral pattern around.

(Figure 1b). Two heterodyne distributed vibration sensing (hDVS) interrogator units from Schlumberger were connected to single-mode fibres and used to record the strain along the boreholes with 5 m spatial sampling. A 20-m gauge length was used for data recordings in the field. Later, this value was adjusted to 40 m for E GrSk 3/90 according to the velocity profile in the reservoir section to get an optimal signal-to-noise ratio and preserve data resolution (Dean et al., 2017). Data recorded in Gt GrSk 4/05 were used with only 20-m gauge length due to the loss of the raw fibre optical data. Although the approach

suggested by Dean et al. (2017) should help to optimise the data quality, a 40-m gauge length might be too high to detect small scale features, such as fractures. Theoretically, a smaller gauge length is necessary for small scale details imaging, such as fractures, but it comes at the expense of having a lower signal-to-noise ratio. More details on gauge length optimisation for the dataset acquired at Groß Schönebeck can be found in Henninges et al. (2021).

The measuring campaign consisted of 1 start-up testing and 3 data acquisition days in total. The 61 vibrator source points

(VPs) had a spiral layout around the target area with varying offsets from 188 to 2036 m around the boreholes to ensure a good azimuth distribution (Figure 1b). Due to budget limitations, the number of vibrator points had to be restricted and often positioned in such a way that they don not have overlapping ray penetrating areas. As seismic source, four heavy Mertz M12

Hemi 48 vibroseis trucks were used with a peak force of 200 kN (45100 Lbf) each. All source units vibrated simultaneously at each VP location and guaranteed an average vertical stacking fold of 16 per source location. A linear sweep of 10 – 112 Hz

and 36 s length was used for data acquisition. Several VPs with larger offsets, were recorded using a sweep from 10 to 96 Hz.

Due to the cable failure in Gt GrSk 4/05, the recording of the last 500 m in the reservoir section was lost, and the maximum surveyed depth is limited to 3716 m measured depth (MD). Moreover, we were only able to record 18 VPs from the originally planned 61 because the cable was retrieved from the borehole after only 1 day of acquisition. This event led to significantly reduced subsurface coverage of the survey design between the two wells. In E GrSk 3/90, we recorded the planned 61 VPs and

the maximum surveyed depth is at 4251 m MD. Nevertheless, recorded datasets from the second borehole have inconsistent amplitudes. This behaviour could be related to the local repositioning of the cable inside the borehole since similar reduced-amplitude patterns were observed in the recordings with extra slack provided to the cable (c.f., Henninges et al. (2021)). Further research is required for a systematic understanding of the here qualitatively explained effects. Overall, all mentioned details make this dataset acquired at the Groß Schönebeck geothermal research site very challenging for data processing. We will

focus on the processing flow in the next section of the paper, which we used to successfully identify reservoir details.

## 3    Data processing

For the 3D DAS VSP dataset a processing flow containing typical elements was adapted to the Groß Schönebeck survey specifics and then applied to the data (see Table 1). The major steps and parameters details are discussed in the following sub chapters.

### 3.1    Data conditioning

As first step the proper geometry was assigned to the raw uncorrelated data, which included source and receiver coordinates, elevations and true vertical depths calculated using boreholes trajectories. Each VP recording set contained a various number of recorded sweeps, ranging from 12 to 37. Data within each VP were sorted on increasing MD and vertically stacked using a trimmed mean stack, which helped to exclude amplitude outliers. Wireline records are frequently suspected to ringing noise,

which represent a standing wave phenomenon, occurring in those depth intervals of the boreholes where the cable can move freely (Martuganova et al., 2021). This type of the noise creates resonances with a fundamental frequency and higher overtones in the amplitude spectrum.

Figure 2 shows selected VPs with variable offsets such as 214 m, 510 m and 1411 m for the borehole E GrSK 3/90 after pre-processing (vertical stacking, correlation with the pilot sweep and subsequent differentiation) and denoising. All seismograms

have a distinct P wave arrival (Figure 2, blue arrows); however, they are heavily dominated by a coherent characteristic striped or zig-zag noise. For instance, depth intervals 904 - 980, 1588 - 1816, 2066-2372 m for VPs recorded in E GrSK 3/90 are contaminated by this type of noise (Figure 2, red arrows panels a, b, c). It appears that the noise distribution does not change significantly from one VP to another and affects shallower depth regions more than the deeper ones. Also, it can be noted that

**Table 1.** 3D DAS VSP data processing flow for wells E GrSk 3/90 and Gt GrSk 4/05.

| Processing step | Description and parameters |
|---|---|
| Geometry input | Source and receiver coordinates, depths |
| Trimmed mean stack | Suppression of impulsive noise |
| Ringing noise suppression | Matching pursuit decomposition (MPD) with Gabor atoms |
| Correlation with pilot sweep | 10 - 112 Hz, 10 - 96 Hz, 36 s, 360 ms taper |
| Conversion to strain rate | Time derivative |
| First arrival time picking | The peak of the direct downgoing P-wave |
| Amplitude corrections | Spherical divergence compensation ($t^{1.5}$) and lateral equalisation |
| Coherency enhancement | Moderate wavefield sharpening by tau-p method |
| Velocity model building | Migration velocities from 3D surface seismics and sonic log from E GrSk 3/90 as initial velocity model, checked and optimised by ray tracing |
| Ray tracing | Mapping of the reflectivity for all source receiver pairs |
| Wavefield separation | Subtraction of downgoing P-wave field (median filter) by a 9 trace median operator |
| Deterministic deconvolution | Waveshaping zero-phasing of the upgoing wavefield, removal of multiples |
| Polarity reversal | 180° phase shift, to match polarity convention of conventional geophone data |
| 3D imaging | 3D Kirchhoff migration with restricted aperture of 12 degrees |

borehole Gt GrSk 4/05 has more noisy intervals, in comparison with E GrSK 3/90, and with higher amplitudes, for example,
for depth regions at 873 - 980, 1697-1848, 2025 - 2177, 2898-3017, 3330 - 3409 m (Figure 2, red arrows panels g, h, i).

Ringing noise clearly represents a challenging problem and should be tackled by means of data processing. We did a few denoising test using different denoising approaches, including Burg adaptive deconvolution (Burg, 1972, 1975) and Time-Frequency domain attenuation (Elboth et al., 2008). The optimal denoising result was achieved using a novel approach based on matching pursuit decomposition (MPD) using Gabor atoms, described in Martuganova et al. (2021). According to this
method we formed an overcomplete Gabor dictionary to decompose the original signal. Then using atomic parameters such as amplitude, frequency and position in time we determined parts of the signal representing slapping of the cable and subtracted them from the data to perform the denoising. To improve the signal-to-noise ratio of the data we applied the MPD denoising method after stacking, but before correlation to avoid smearing the noise by the correlation process. The results of denoising for two boreholes are given in Figure 2 on panels d, e, f and j, k, l for E GrSK 3/90 and Gt GrSk 4/05 respectively. MPD
denoising eliminated almost all noise on the seismograms and significantly improved traceability of the reflections (Figure 2, green arrows panels d, e, f and j, k, l), which are no longer obscured by the ringing noise.

After denoising, the correlation with the pilot sweep and time-differentiation to convert data to the strain-rate were applied to the data. This was followed by amplitude corrections (spherical divergence correction and lateral equalisation) and moderate

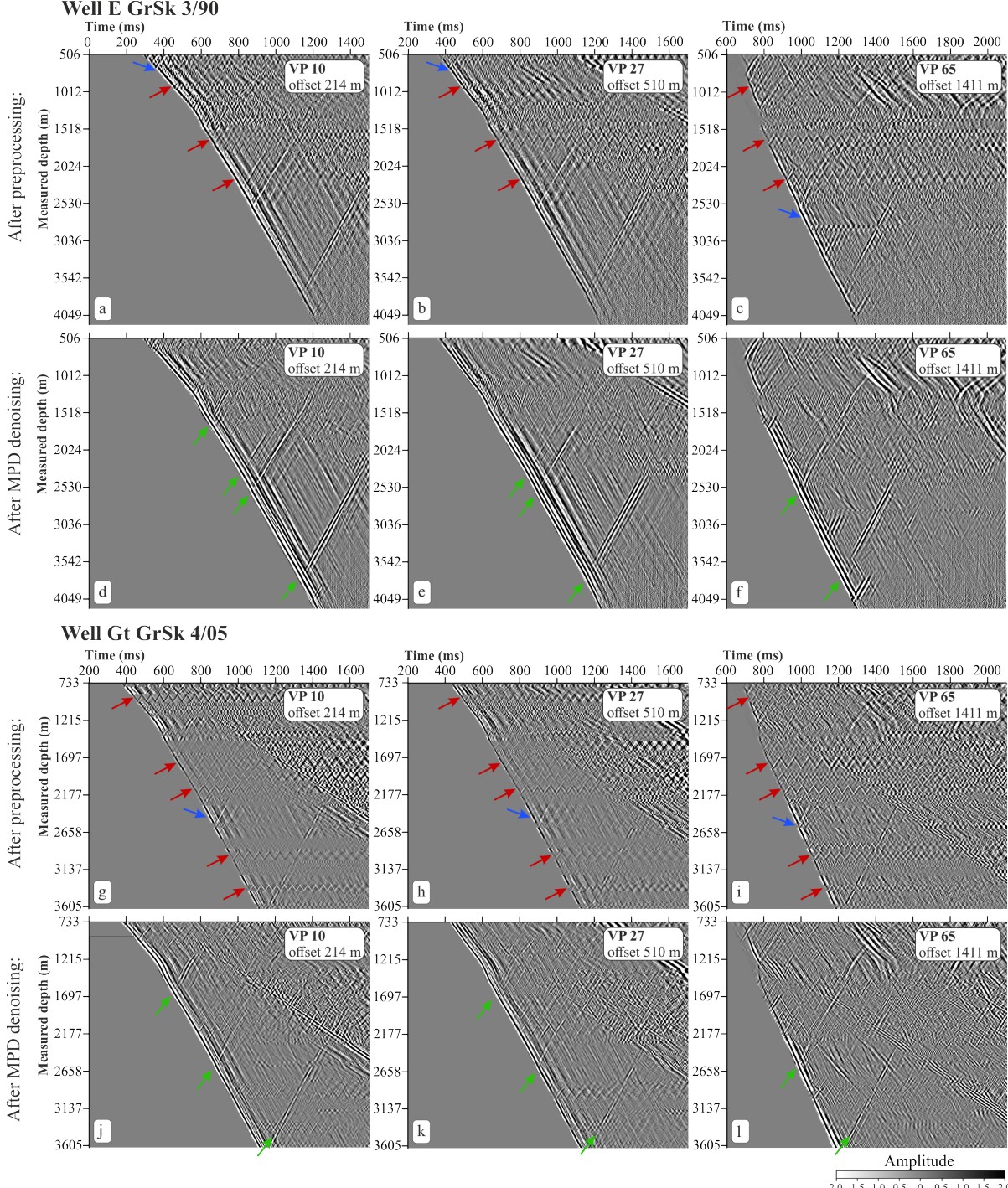

**Figure 2.** Common-source gathers displays for different source positions with offsets 214 m, 510 m and 1411 m after preprossessing (a), (b), (c) for well E GrSk3/90 and (g), (h), (i) for well Gt GrSk4/05; after noise subtraction using MPD denoising method and a moderate coherency enhancement (d), (e),(f) for well E GrSk3/90 and (j), (k), (l) for well Gt GrSk4/05. Arrows are colour-coded as follows: direct downgoing P wave (blue), upgoing reflected P–P waves (green), and intervals affected by ringing noise (red).

wavefield enhancement. Also, measured depths were converted to true vertical depth below seismic datum elevation (TVDSD), and later in the text, we refer to it as depth.

## 3.2 Velocity model building and ray tracing

Several independent data sources were used to determine the velocity function for main stratigraphic layers of the region and assign appropriate P-wave velocities (Figure 3). First, velocity profiles recorded at zero offset source position were calculated with the Lizarralde smooth inversion method (Lizarralde and Swift, 1999) to get the main trend (Figure 3a black curve). Then, velocity values were updated according to calibrated sonic log data (Figure 3a thin grey curve) and the surface seismics velocity cube (Figure 3a, dark blue curve). The model was checked using the ray tracing results calculated in commercial software VSProwess X (VSProwess Ltd.), and iteratively optimised to minimize the drift between the recorded and modelled arrival times for the rig shot at source position 10.

To improve the fit for far offset VPs a small anisotropic drift was included into the model. Thomsen's P-wave anisotropy parameters (Thomsen, 1986), namely $\epsilon$ and $\delta$, for transversely isotropic (TI) media, 4 % $\delta$ and 16 % $\epsilon$ were selected for all layers down to the top salt. These parameters were tested and optimised only using DAS VSP data, which allowed to reduce the standard deviation of drift for the longer offset VPs. Finally, all VPs were ray-traced through the constructed model, and reflection points (loci) for each source-receiver pair were extracted. The picked arrival times were compared with the ray traced times to check whether the smallest misfit for all available data was achieved. The average of the mean model drift for all VPs is 2.43 ms for the borehole E GrSk3/90 and 7.7 ms for Gt GrSk 4/05, respectively.

The layered 3D DAS VSP velocity model follows the geological model (Moeck et al., 2009) and has constant layer velocities or vertical velocity gradients indicated by the sonic log data (Figure 3a red curve; Figure 3b). Potential lateral variations are not accounted for. The values vary from 1750 m/s in the upper Quaternary and tertiary layers to 5000 m/s in the Rotliegend (Permian) reservoir section.

## 3.3 Data preparation for migration

The next processing step includes wavefield separation and deconvolution. To separate the upgoing wavefield the downgoing P-wave field was subtracted using a median filter. We tested a few different wavefield separation techniques (FK transform, FP transform), and the cleanest result was obtained by applying the median filter. This filter delivered a cleaner residual upgoing response with less smearing of the amplitude artefacts, and also preserves the resolution of data. Prior to deconvolution, the upgoing wavefields were scaled by -1 changing the polarity of DAS data to match the required European convention (increase AI = negative number on a trace.) The separated wavefield was deterministically deconvolved, using individual downgoing responses as trace-by-trace operators, followed by a bandpass filter 8, 12 – 70, 80 Hz and a front mute.

To further prepare the data for imaging, source static corrections were applied. Additionally, data were moved to pre-migration depth using a model-based stretch to map points according to reflection-point loci. Due to significant data quality variations between shots caused by unknown changes in the response of the fibre, the amplitudes across all data points were normalised using RMS amplitude values calculated in a depth window selected on the most prominent salt sequence reflection

event. Each trace then was scaled by the inversed RMS trace value. The resulting pre-migrated seismograms were used as input for 3D imaging.

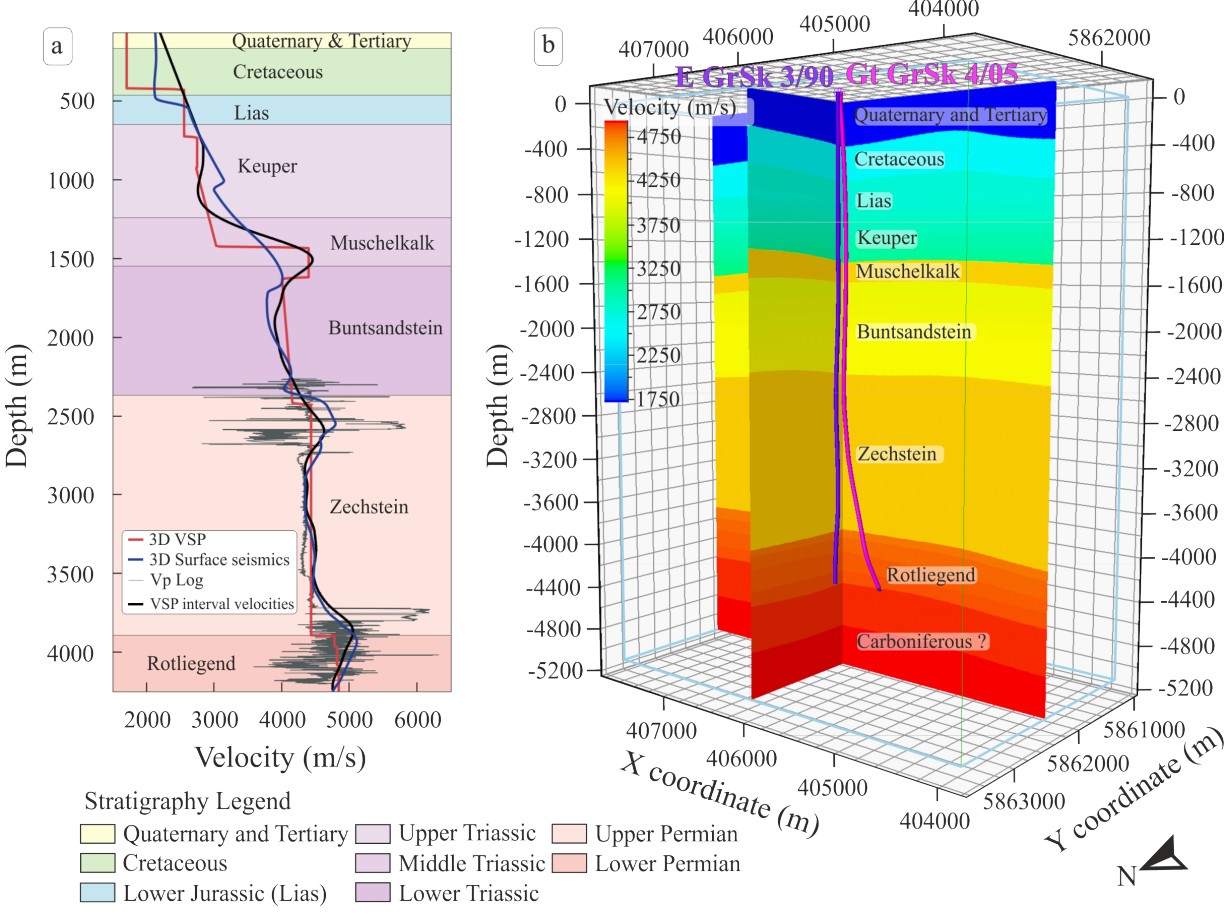

**Figure 3.** Velocity information at the Groß Schönebeck research site. (a) Velocity functions at the borehole E GrSk3/90 location. Red line - the curve extracted from the 3D DAS VSP velocity model shown in (b); dark blue line - the curve extracted from the 3D velocity model used for the 3D surface seismics prestack depth migration; thin grey line - sonic log measurement; thick black line - VSP interval velocities derived from the first break peaks using the method of smooth inversion after Lizarralde and Swift (1999). (b) 3D DAS VSP velocity model created using top formation surfaces to build an initial velocity model, which was then checked and optimised according to ray tracing results.

### 3.4 3D imaging

For imaging, a commercial 3D Kirchhoff migration algorithm (VSProwess Ltd.) was used. Calculated via ray tracing, the reflection-point loci for each source-receiver pair are interpolated and used to map each processed sample to its modelled image point coordinate. This is routinely known as VSP common depth point (CDP) mapping (Dillon and Thomson, 1984).

Once the reflectivity is mapped approximately to the correct location it can be binned. An efficient surface tracking algorithm is used to find all those bins within the specified aperture angle intersected by the reflection ellipsoid.

For migration, we used a 12.5 m x 12.5 m horizontal and a 5 m depth bin size. After extensive testing, we chose a 12-degree aperture which allowed to sum the main reflections best and to preserve image details. In addition, the migration operator

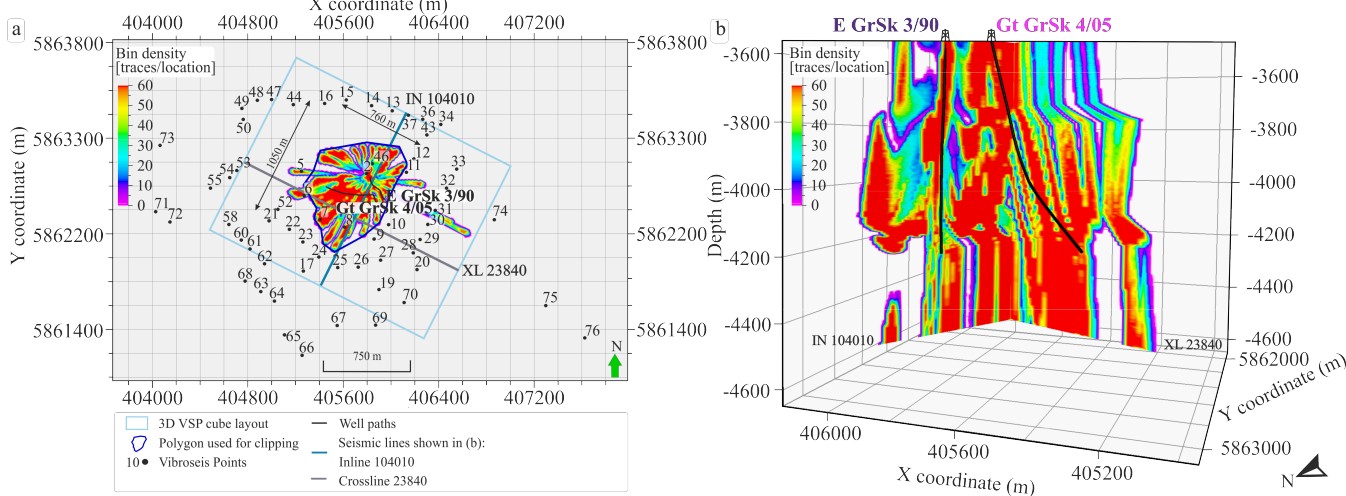

**Figure 4.** Examples of the ray coverage for the study area. Depth slice of a 3D bin density volume at 4100 m depth (a). 3D visualisation of two seismic lines in the 3D bin density cube (b) (for location see a).

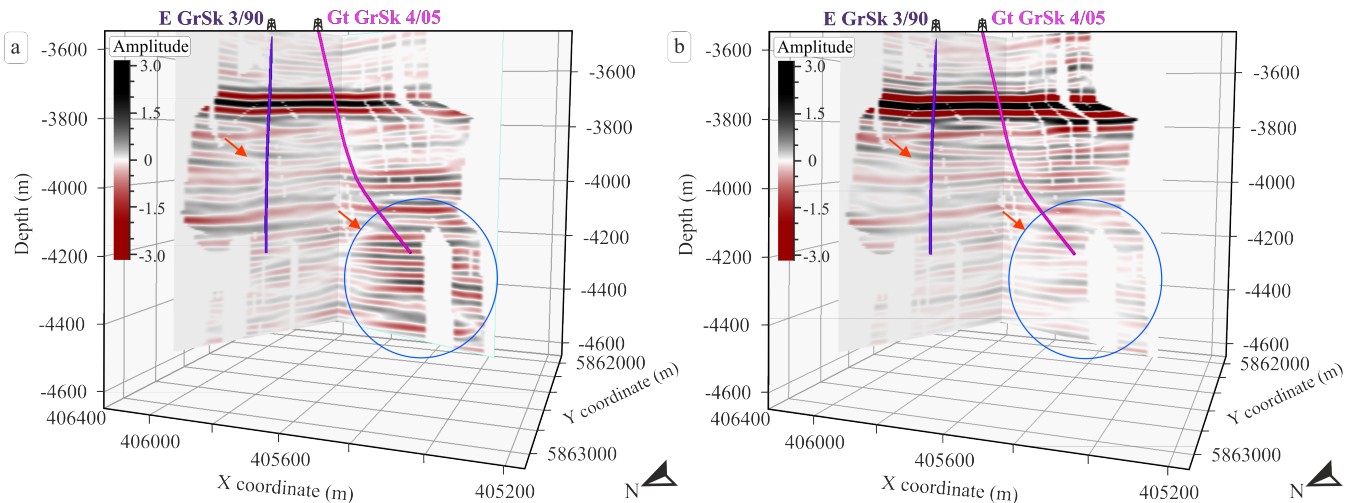

**Figure 5.** A comparison of 3D DAS VSP cubes generated with a model-based Kirchhoff migration algorithm (a) without and (b) with MPD denoising.

removed some of the imaging artefacts. A scaled version of each mapped sample is summed into each of these bins (normally cos-squared window). A by-product of migration is a bin count cube. A 3D bin density image along two lines and the depth slice at 4100 m are shown in Figure 4. This 3D visualisation discloses holes in the seismic coverage at 4100 m depth. Gaps in coverage are particularly noticeable east of the observation well. To avoid imaging artefacts, bin density information was used first to normalise amplitudes in the 3D DAS VSP volume. Afterwards, the bin density information is utilized to identify high uncertainty areas where the image fold is sparse and to clip the resulting cube in accordance to it. Furthermore, the bin density cube was employed to determine an area with a reasonable coverage (Figure 4a, dark blue polygon) for further clipping of horizon maps.

The suppression of the ringing noise by means of MPD denoising and amplitude normalisation techniques, significantly improved the imaging results. Figure 5a shows the result of the migration which excludes denoising from the processing flow. The first problem that can be noticed are inconsistent significant amplitude anomalies, especially clearly visible around borehole Gt GrSk 4/05 (Figure 5, blue ellipse) in the reservoir section from 4000 m till 4500 m depth. Apart from that, the "noisy" cube has decreased resolution and horizon continuity in comparison with the "clean" cube (Figure 5, orange arrows).

## 4 Reservoir imaging

The resulting 3D DAS VSP cube image is 1600 m x 2000 m, which has a relatively limited illumination range with a maximum extent of 760 and 1050 m along inline and crossline direction respectively (extension estimated based on polygon for clipping; Figure 4). The image of the subsurface is the most complete and best around the boreholes. Reflections in the vicinity of the wells reach up to 4500 m depth (Figure 6). To interpret the 3D DAS VSP imaging results (Figure 6a), we compare it with the 3D surface seismic cube (Krawczyk et al., 2019; Norden et al., in revision) after prestack depth migration (Figure 6b). The polarity of the DAS data was changed to match the polarity of the geophones. The 3D surface seismics was restricted according to the 3D DAS VSP cube layout. Bin size for borehole volume is 12.5 m x 12.5 m and for surface seismic volume is 25 m x 25 m. This means that inlines / crosslines of the 3D DAS VSP cube are 2 times denser.

The reservoir section is situated at a depth interval between 4 - 4.5 km and has an average velocity around 4700 m/s; the dominant frequencies in 3D surface seismics are between 25 and 47 Hz. With wireline 3D DAS VSP the frequencies are from 34 to 73 Hz. Vertical seismic resolution can be estimated as quarter of the wavelength ($\lambda$), which depends on velocity ($V$) and frequency ($F$) as follows:

$$\lambda = \frac{V}{F}. \tag{1}$$

This gives a vertical resolution estimation of 25 - 47 m for the conventional surface seismic cube and a more favourable 16 - 34 m for the borehole seismic dataset.

Several reflectors can be mapped with high confidence across both volumes. Especially the main marker horizons of the typical stratigraphy in the North German Basin were successfully imaged (Figure 6): top Staßfurt-Basalanhydrit Fm ($Z_1$), top Rotliegend ($Z_3$), within Mellin-Schlichten ($R_1$), within Dethlingen Fm ($R_3$), the top and the bottom of Elbe sandstone reservoir

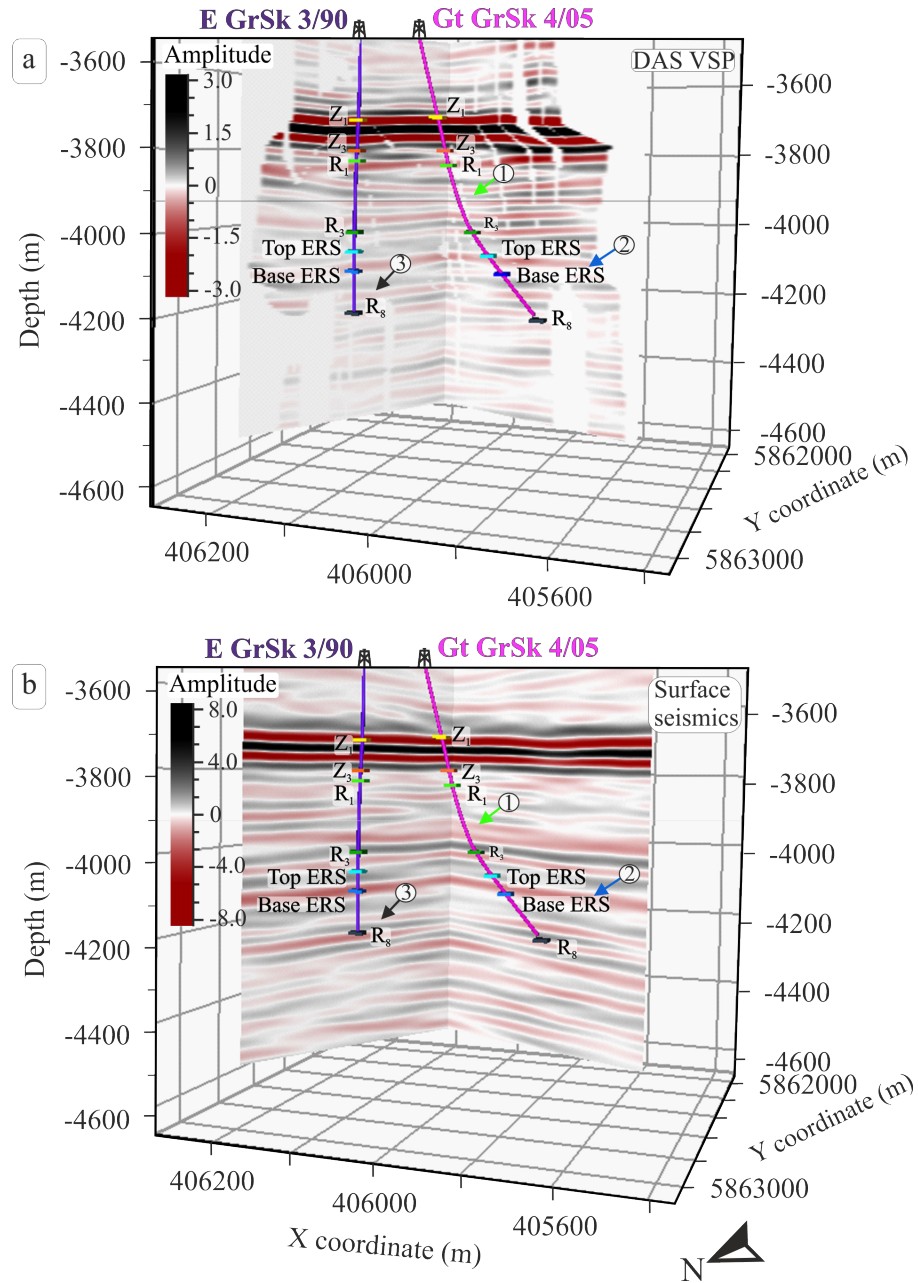

**Figure 6.** A comparison of 3D DAS VSP cube (a) and prestack depth migrated 3D surface seismic cube (b). Arrows with numbers highlight major interpretation features in the 3D DAS VSP and 3D surface seismic cubes. Light green arrows (1) point out the difference in seismic resolution in the upper Rotliegend; dark blue arrows (2) show the position of the Elbe reservoir sandstone horizon. Dark grey arrows (3) mark the location of unconformity in lower Rotliegend formation. Main seismic reflectors labelled on the figures: top Staßfurt-Basalanhydrit Fm ($Z_1$), top Rotliegend ($Z_3$), within Mellin-Schlichten ($R_1$), within Dethlingen Fm ($R_3$), top ESR - top of Elbe sandstone reservoir, base ESR - base of Elbe sandstone reservoir, $R_8$ - Base Effusive Rotliegend.

(top ESR, base ESR), and possibly top of the Carboniferous ($R_8$). Although both cubes have similarities, there are distinct
differences related to the higher vertical resolution of the VSP measurements. In the following chapters we will focus on each
structural feature separately.

## 4.1 Upper Rotliegend horizons

The most prominent reflections in both seismic cubes are closely situated reflection bands from the transition from salt to
anhydrite (Staßfurt-Basalanhydrit Fm ($Z_1$)), followed by the reflection $Z_3$ from the base of Zechstein. Together these closely
situated seismic responses create a complex wavelet superimposition consisting out of 5 phases (Figure 6). Most clearly a
characteristic Zechstein reflections are visible on the Figure 7, which shows a seismic cross-section extracted between two
boreholes with well logs (gamma ray (GR), bulk density (RHOB) and sonic velocity (Vp)), lithology data and stratigraphy.
Salt layers are underlayed by upper Rotliegend sediments.

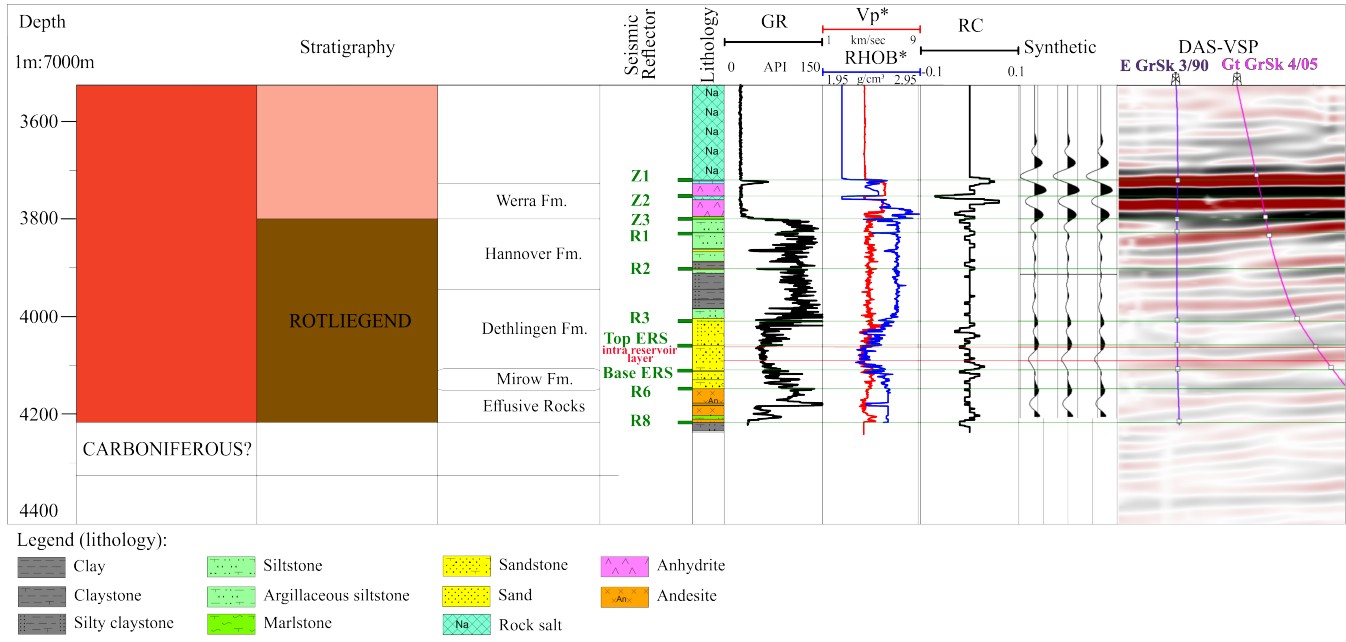

**Figure 7.** DAS-VSP seismic cross-section through the boreholes plotted together with well logs (GR: gamma ray, Vp: sonic velocity, RHOB:
bulk density), lithology, stratigraphy, reflector coefficient (RC) and synthetics.

Göthel (2016) refer to seismic horizons in Rotliegend as "Phantom horizons", since they are highly variable in depending
on regional geological settings and do not have a reliable definition. For the first time in the area of the research platform Groß
Schönebeck, borehole seismics allows the separation of thin interlaying of siltstone and silty mudstone structures in the upper
Rotliegend sediments in the depth range from 3800 m to 4000 m (Figure 6a, light green arrow (1)). On the 3D surface seismics
(Krawczyk et al., 2019; Norden et al., in revision) a thick unresolved high amplitude continuous layer with visible thickness
variations can be observed at approximately 3900 m (Figure 6b, light green arrow (1)). In contrast to the 3D surface seismic

cube, on the 3D DAS VSP cube, two closely situated thin layers can be traced with a relatively constant thicknesses (Figure 6a, light green arrow (1)).

By comparing the depth sections between 3800 m and 4000 m on both cubes, it can be noted that thin interlaying horizons in the 3D DAS VSP cube have various dipping characteristics, whereas the 3D surface seismic cube shows only thicker horizontal lines, sometimes even not continuous and with amplitudes variations along them. This might be related to the difference in the frequency content of surface and borehole seismic surveys, and as a result with the latter case having a higher resolution. The results of 3D DAS VSP imaging allows to trace a few thin horizons in the upper Rotliegend interval.

Figure 8 demonstrates an inline and a crossline extracted from the central part of the seismic cube with interpreted horizons corresponding to reflections within Mellin-Schlichten ($R_1$) and within Dethlingen Fm ($R_3$). A negative amplitude event associated with a sandstone interlayer inside siltstone sediments of Hannover Fm around the depth of 3800 m could be followed throughout the DAS-VSP volume and is interpreted as reflector $R_1$ (Göthel, 2016; Moeck et al., 2009). There is a decrease in gamma-ray, neutron porosity and Vp logs at this interval, resulting in decreased acoustic impedance values (Figure 7). A positive amplitude event at a depth of 4010 m is mapped as the reflector $R3_3$ (Henninges et al., 2021; Norden et al., in revision). It marks the transition from the Dethlingen sandstones to a succession of siltstones, followed by mudstones. A change in lithology can be identified by increased gamma ray, density, and sonic velocity values, which leads to the increase of acoustic impedance (Figure 7). Depth variations of these horizons are visible already in a small volume around the boreholes. The indication of these variations in lithological sequences is also present in the corridor stacks within this interval (Henninges et al., 2021).

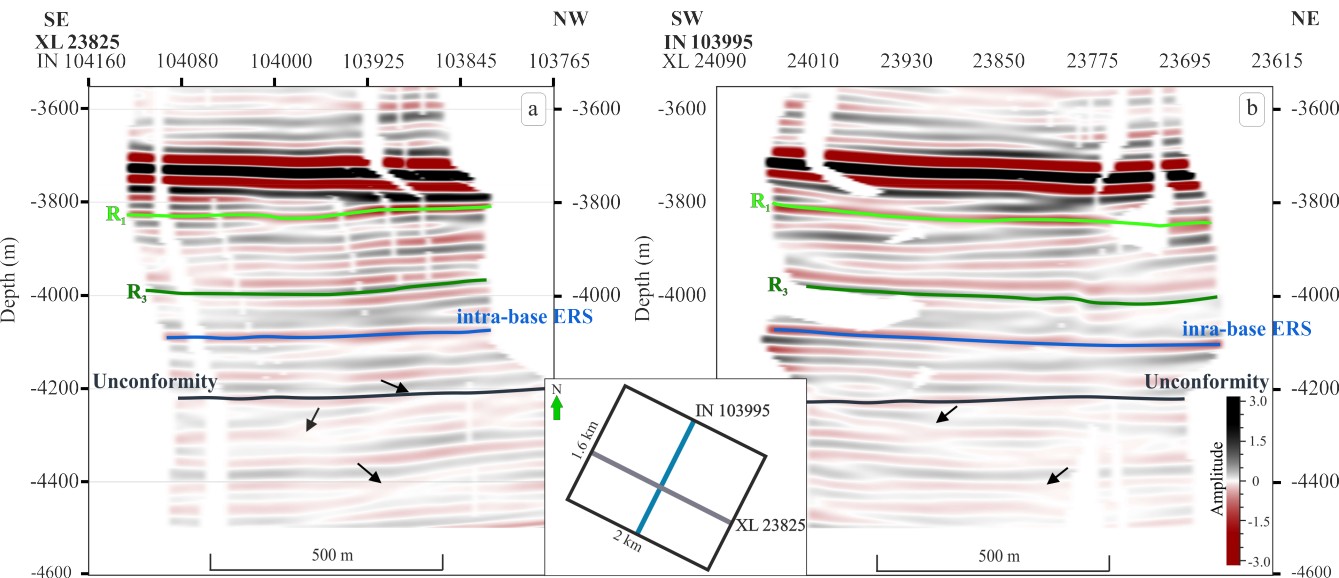

**Figure 8.** Seismic lines extracted from the 3D DAS VSP cube with horizons interpretation. (a) Seismic cross-line and (b) seismic inline, showing thin interlayered horizons in the upper Rotliegend (light green and dark green lines), base of Elbe sandstone reservoir (dark blue line) and the Lower Rotliegend unconformity (dark grey line). Black small arrows point out the locations of the numerous pinch outs.

## 4.2 Elbe reservoir sandstone layer

One of the possible targets for geothermal exploration are sandstones of the Dethlingen formation / Lower Elbe subgroup. In the Brandenburg area the lower part of the Dethlingen formation comprises fine-to-coarse grained sandstone with high quality reservoir properties (porosity 8-10 % and permeability of 10 - 100 mD (Trautwein and Huenges, 2005)). This layer was deposited in aeolian setting and then reworked by aquatic processes. The Elbe reservoir sandstone (ERS) layer is located between 4060 to 4100 m depth within the sandy section of the Dethlingen Formation in the E GrSk 3/90 well (Bauer et al., 2020) (Figure 7). On well logs this interval is characterised by decreased P-wave velocity, caused by an increased porosity of this section (Trautwein and Huenges, 2005). This geological unit was successfully imaged on both 3D images (Figure 6, dark blue arrows (2)).

In the conventional surface seismic image (Krawczyk et al., 2019, Norden et al., in revision), the base of ERS horizon can be traced as a continuous negative phase at around 4080 m depth, with increasing thickness toward the southwest direction. Even the theoretical resolution should be between 24 - 47 m. Bauer et al. (2020) showed via finite-difference forward modelling that this complex layer in the reservoir section still won't be adequately resolved since the theoretical resolution cannot be achieved due to challenging geological settings above and below the ERS.

Within the depth range from 4060 to 4100 m the depth section from 4070 to 4090 m shows a low variability in logs values, indicating an even "cleaner" part of the ESR sandstone layer (Figure 7). The high resolution of the 3D DAS VSP cube allows seeing an internal structure inside this interval and tracing depth variations of the base of the intra-reservoir horizon within Elbe sandstone reservoir. It may represent porous parts of a stacked fluvial sandstone body within the sandy Dethlingen Fm. succession. Below in the text we will refer to this horizon as intra-base of Elbe reservoir sandstone layer (intra-base ERS). This interval's base is characterised by negative phases (decreased acoustic impedance) on the seismic 3D DAS VSP cube. We picked this horizon through the entire volume and created a depth contour map (Figure 9a). The map was clipped using the dark blue polygon shown on Figure 4a, to avoid interpolation artefacts in the regions with low coverage. The intra-base ERS horizon lies at 4080 m on the southwest and at approximately 4100 m depth on the northeast. On the seismic section (Figure 9b), it is clearly visible that there is a pinch out on the southwest part of the profile, which was not distinguishable on the 3D surface seismics cube and visible as a thickness variation.

The top of the ERS horizon corresponds to positive phase (increased acoustic impedance) on 3D DAS VSP cube. In general it follows similar paleo relief of the intra-base ERS with the deepest values around 4050 m in the southwest and 4080 m in the northeast (Figure 10a). However, local depth variations are present, and therefore the thickness of the Elbe reservoir sandstone layer is highly variable in close proximity to the boreholes (Figure 10b). It ranges from 20 to 35 m.

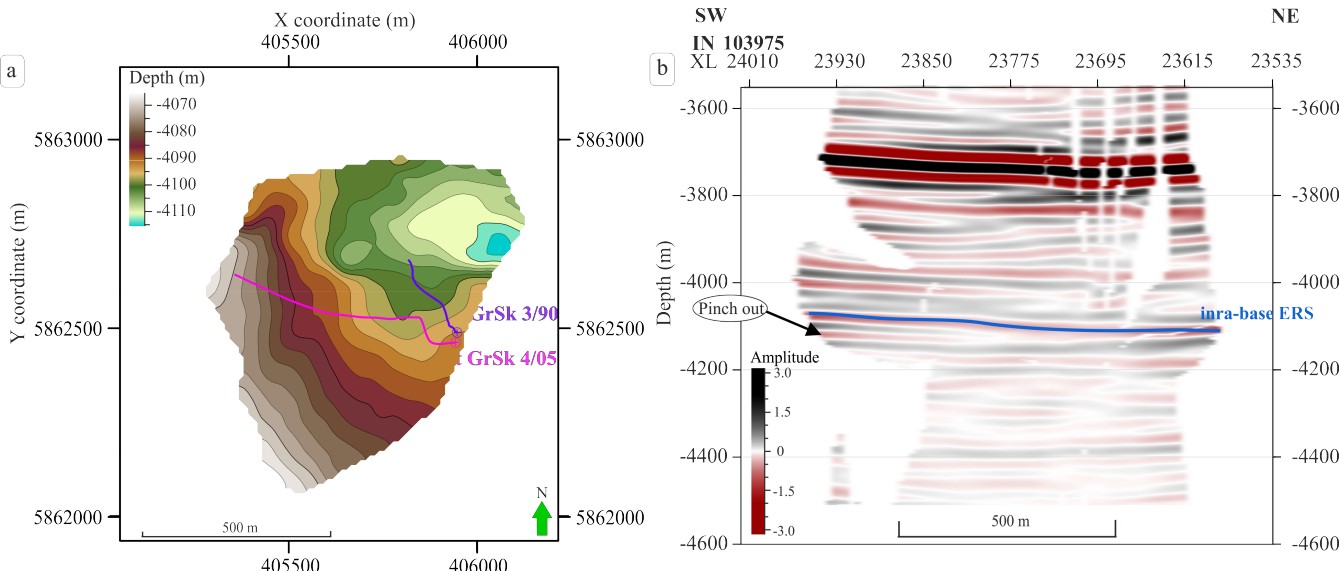

**Figure 9.** (a) Spatial extent of the Elbe reservoir sandstone layer. Depth contour map of the Elbe sandstone reservoir intra-base horizon. The dark blue line marks the location of the inline shown in (b): seismic section from the 3D DAS VSP volume with the interpretation of the internal structure of the Elbe reservoir sandstone (ERS) horizon.

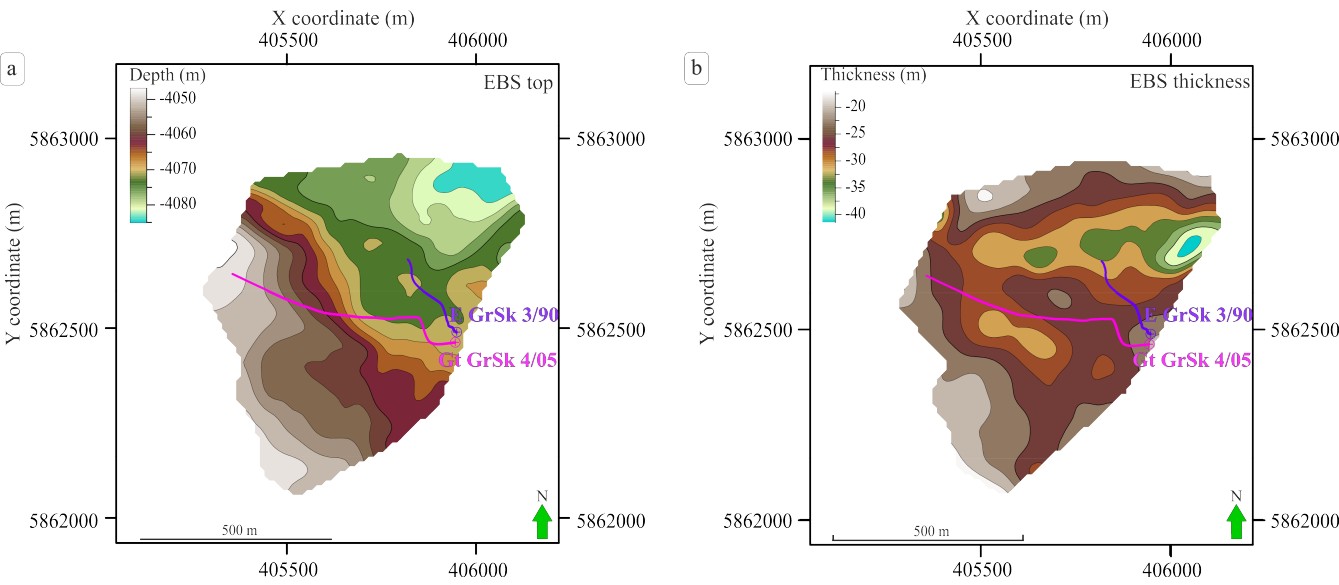

**Figure 10.** Mapping results within the study area: depth contour map of the top Elbe reservoir sandstone layer (a) and the ERS thickness (b).

## 4.3 The lower Rotliegend unconfomity

Another essential feature of the seismic interpretation on the 3D DAS VSP and the 3D surface seismics (Krawczyk et al., 2019; Norden et al., in revision) cubes is the change in seismic reflection pattern (Figure 6) at a depth around 4200 m. On the 3D surface seismics, this is visible as a change from horizontal continuous reflectors to layers with inconsistent amplitudes and a
lower reflectivity in the lower Rotlieged formation (Figure 6b, the dark grey arrow (3)). On the 3D DAS VSP cube, a change from nearly horizontal to inclined reflectors can be detected (Figure 6a, the dark grey arrow (3)). This change of the seismic facies evidences the existence of an unconformity in the area where we expect volcanic rocks. We will refer to it as lower Rotliegend unconfomity.

On the seismic cross-line and inline shown in Figure 8 a and b, the possible unconformity boundary is marked by a thick
dark grey line. This reflection has weak, uncertain characteristics. Small black arrows indicate numerous exciting pinch outs below this horizon, which were used as indicators of the type change of the layering. Due to reduced reflectivity and gaps in the cubes, tracing this horizon accurately is a pretty challenging task. Therefore the resulting depth contour map might have errors, especially at the edges of the dataset. Nevertheless, in the area between the two boreholes where we have the best coverage, we can see that the depth variation of the lower Rotliegend unconformity is relatively limited to the depth range between 4200
300    - 4230 m (Figure 11).

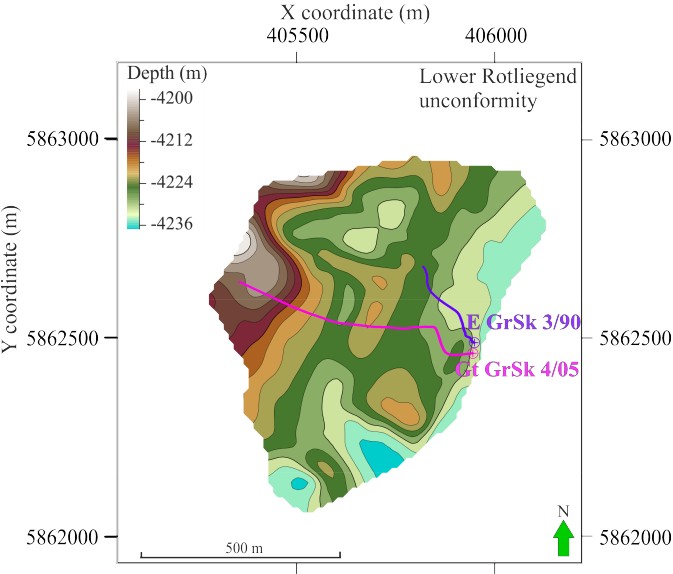

**Figure 11.** Spatial distribution of the unconformity boundary. Depth of the Lower Rotliegend unconformity.

## 5 Discussion & Outlook

### 5.1 The Groß Schönebeck experiment

With this study, we investigated the applicability of the DAS wireline acquisition method for detailed deep geothermal reservoir imaging and the capability to bridge the gap between well logs and 3D surface seismics. 3D DAS VSP provides, on average, a two times better-resolved image than 3D surface seismics within this project and has a significant potential in the geothermal sector. Data acquisition using engineered fibres or new interrogators with higher signal-to-noise ratio can help notably improve the quality of the wireline DAS data recording and compensate for signal loses in the deep reservoir section. Nethertheless, more research on fibre coupling improvement and location control in the borehole is strongly advised, which could allow wireline DAS acquisition to become a routine for numerous applications.

VSP surveys conducted with distributed sensors permanently installed behind casing or along the tubing provide the advantage of good coupling conditions and the possibility to perform time-lapse studies. Successfully reported case studies include applications for unconventional exploration in Texas (Shultz and Simmons, 2019) to create a detailed image of the formation around the well. Application of DAS for CO2 monitoring (Götz et al., 2018; Correa et al., 2019) is a well-known routine to provide a reliable method for targeted, detailed imaging and 4D monitoring of the site. 3D DAS VSP surveys in oil and gas exploration reduce exploration risks in regions with a challenging environment, for instance, in the presence of strong production noise (Jiang et al., 2016), in areas with complex salt tectonics (Bartels et al., 2015) etc. In general, all the surveys with permanently installed fibre optic cable have a better signal-to-noise ratio than data typically recorded with wireline DAS acquisition.

Wireline DAS, however, has a great advantage. It allows to acquire seismic data in already drilled and cased boreholes. This can be done at a low cost and in a small amount of time in comparison with VSP acquired using geophones. Even though wireline 3D DAS VSP at the Groß Schönebeck site has worse coupling conditions than conventional DAS cemented behind the casing, the resolution of the imaged seismic volume is still in the same range of 20 m (Götz et al., 2018; Correa et al., 2019).

To the best of the authors knowledge, there are only limited examples of wireline DAS applications with active seismic sources for geothermal exploration. One known example was recorded in 2016 at the geothermal field at Brady Hot Springs, Nevada (Miller et al., 2018). A fibre optical cable was deployed into the accessible 363 m portion of the vertical well 56-1. This dataset has a limited depth range (up to 297 m only), and only two shot points were used for data acquisition. Trainor-Guitton et al. (2018) presented 3D imaging results from the same geothermal field, which are limited to the depth up to 600 m. Although some reflectors were imaged, they have a Hyperbola-shaped reflections often dominate these imaging results, making the interpretation challenging. Therefore, the experiments conducted at the Groß Schönebeck in-situ laboratory and Brady Hot Springs are different and it is challenging to compare these two datasets. Thus, recorded data at the Groß Schönebeck is unique among the experiments conducted for geothermal exploration, with reflection information recorded down to 4.2 km deep.

A spiral survey design at the Groß Schönebeck site led to a ray focusing on the target reservoir area yielding an equably distributed offset and azimuth coverage. A detailed imaging of the target reservoir interval (with vertical resolution up to 16

335    m) was achieved with only 61 VPs. Nevertheless, low to zero ray coverage in some regions of the 3D VSP survey resulted in a lack of data required for a successful migration process without excessive artefacts. Limited number of vibrator points resulted in a petal-shaped footprints, with the highest coverage in the centre, are prominent in bin density slices (Figure 4a) revealing a partial illumination problem. Consequently, the resulting migrated depth slices have gaps and are not well suitable for classical attribute analysis. These limitations led to the usage of the 3D DAS VSP cube for structural interpretation only.

A more similar experiment to the Groß Schönebeck survey was conducted for hydrocarbons exploration in China's Heibei region (Yu et al., 2016). DAS walkaway and walkaround VSP with wireline survey consisted of 386 successful shot points, 6 times more than at the Groß Schönebeck experiment. The data acquisition for the in-situ laboratory in the Brandenburg area was likely less time-consuming, and cheaper than this experiment in China. However, densely regularly placed source positions allowed to get a detailed resolved 3D image for the area of interest in Heibei region without gaps and prominent migration

artefacts. Most recently Lim et al. (2020) showed a remarkable dataset from methane hydrate research test well in North Slope, Alaska, USA. A survey consisted from 1701 VP arranged around the borehole and a permanantly installed fible optic cable. Phenomenal dataset quality of the 3D DAS VSP data revealed indications of the sub-fault system that are not presented in the surface seismic data. The experiment in China and Alaska showed that with a larger amount of source points a better coverage can be achieved, however, one should always look for a trade off between a reasonable coverage and acquisition cost.

**5.2   Data processing**

The experiment at the Groß Schönebeck allowed gaining valuable knowledge on survey planning and data processing. One of the biggest challenges for this dataset was a ringing noise problem. At the early stages of the data processing, an intermediate solution for denoising included Burg adaptive deconvolution combined with careful exclusion of depth intervals with a poor signal-to-noise ratio from the data processing. To further improve imaging results and limit migration artefacts due to sparse

coverage, it was necessary to improve the signal-to-noise ratio of the data and include more data into migration calculations. Therefore, an essential part in successful data imaging results played careful denoising using the MPD approach.

Besides the quantity of the data, another important parameter, which will significantly influence the imaging results, is the migration aperture (Schleicher et al., 1997; Sun, 2000). After extensive testing we came to the conclusion that a strict restriction of $5°$ can lead to an ambiguous and inconsistent summation of the main horizons for our dataset. On the other hand,

a too broad aperture can reduce the resolution of the horizons. Thus, a compromise between these parameters should be found. Additionally, due to its stacking nature, the migration operator will also attenuate to some extent the residual noise in the data.

**5.3   Future geothermal exploration plans**

The results of our 3D DAS VSP experiments prove that wireline DAS VSP measurements can significantly contribute to exploration campaigns. The wireline DAS VSP allows reducing risks and cost, and can have a higher resolution compared to

the conventional 3D surface seismics interpretation which does not provide enough information due to the limited resolution of the data. This case study can be of special interest for geothermal wells with complex structures, or with thin reservoirs which are hard to image. Especially deep thin geothermal reservoirs, or reservoirs that require stimulation with a low economical

value may benefit from wireline DAS, for which a conventional VSP would otherwise have been cost prohibited. These deep reservoirs may require a high frequency content for high resolution imaging which is only preserved with a VSP due to the one way travel path.

The Elbe reservoir sandstone layer currently represents one of potential targets for the future geothermal exploration. The 3D VSP imaging results clarified the sandstone layer's effective thickness with good reservoir properties. According to our estimations, it varies between 25 to 40 m (Figure 10b) near the boreholes locations. The fluvial nature of these deposits is perhaps responsible for this high variability. Previously the effective thickness of the Elbe reservoir layer was estimated at around 80 m (Zimmermann et al., 2010). The updated thicknesses from the 3D surface seismic experiment, calculated using wavelet transform-based seismic facies classification, showed the predominate thickness of 40 m (Bauer et al., 2020; Norden et al., in revision). Our imaging results from the 3D DAS VSP further refine the reservoir geometry and reveal thickness estimations of the intra-reservoir layer.

The observed lower thicknesses may explain why a matrix-dominated exploration approach did not succeed at the Groß Schönebeck as the reservoir volume which is able to contribute to the fluid flow is considerably lower than expected beforehand. However, the mapped variations of the more porous reservoir thickness, representing most likely the variability of facies-related petrophysical properties, should be considered for the design of the fracture-dominated geothermal systems. Further investigations on fluid flow estimations should be conducted using independent temperature data, well logs and hydraulic tests data.

At the depth level around 4200 m, we mapped the lower Rotliegend unconformity. There is a hypothesis that deposits of permo-carboniferous volcanic rocks lay below this border, which represent another possible target for future exploration (Norden et al., in revision). According to literature sources (Guterch et al., 2010), lower Rotliegned volcanics have a significant time-gap in sedimentation and therefore mapped unconformity horizon could indicate a difference in layering caused by erosion. The 3D DAS VSP imaging successfully contributed to the determination of this critical boundary since it was not well characterised on the 3D surface seismics cube. The bottom of permo-carboniferous volcanic rocks is not detectable neither on the 3D DAS VSP nor on the 3D surface seismic cubes. This implies that the thickness of this deposit's sections can be greater than 300 m. Considering this information, the economic profitability is significantly higher for treated volcanic rocks than for the Elbe reservoir sandstone layer. Nevertheless, we believe it is crucial to determine the exact depth by drilling, perform core analysis, and use well logging methods to determine the precise composition of sediments below this border and essential parameters such as porosity and permeability before developing concepts for possible reservoir treatments in volcanites.

The exact development plans of the site are still under discussion. Possible scenarios include implementing a new stimulation concept and possibly drilling a new well (GrSk 5) or deepening the existing borehole E GrSk 3/90.

## 6 Conclusions

We analysed the 3D DAS VSP imaging results acquired with wireline DAS installation at the Groß Schönebeck geothermal research site. Despite the numerous difficulties during the data acquisition campaign, the borehole seismics was able to image the target interval and substantially contribute to the detailed interpretation of the geothermal reservoir.

The 3-dimensional image created using borehole yields:

- Resolution of thin complex upper Rotliegent geological structure;

- Mapping of the Elbe reservoir sandstone;

- Detection of the lower Rotliegend unconformity in the region with possible volcanic rocks.

The interpretation of the 3D DAS VSP cube evidenced the unexpected absence of faults with larger vertical offsets and fractures. Furthermore, no indications for free gas were found in the data. These findings are important for the further development of the Groß Schönebeck in-situ geothermal laboratory. Wireline DAS allows for a significantly increase in the number of sensors and a larger recording aperture, which results in imaging over a larger depth interval. Thus, it is cheaper and faster than the application of conventional borehole sensors. Nevertheless, careful survey planning and sophisticated data processing are vital for successful imaging results. This successful case study at the Groß Schönebeck site can play a crucial role in developing and applying modern, efficient geothermal exploration methods in the Northern German Basin and other regions with comparable lithology.

*Data availability.* Data will be available after the embargo period at the end of the year via the GFZ repository (dataservices.gfz-potsdam.de/portal).

*Author contributions.* JH, MS and CMK planned the experiment. JH and MS supervised the fieldwork and data acquisition. EM performed the seismic data processing and analysis under supervision of MS and CMK. EM interpreted the data under discussion with with all co-authors. EM, MS, BN, JH and CMK discussed the results and contributed to the final manuscript. JH and CMK supervised the project.

*Competing interests.* The authors declare that they have no conflict of interest. Charlotte M. Krawczyk is chief executive editor of SE.

*Acknowledgements.* Funding of the present work was provided by the German Federal Ministry of Economic Affairs and Energy (grant 0324065), and the European Commission, Horizon 2020 Framework Programme (grant nos. DESTRESS (691728) and EPOS IP (676564)). The publication of this work is supported within the funding programme "Open Access Publikationskosten" Deutsche Forschungsgemeinschaft (DFG, German Research Foundation) - Project Number 491075472. We would like to thank all contractors involved in the acquisition and processing. The authors would like sincerely to acknowledge Ernst Huenges for establishing research activities at the Groß Schönebeck

in-situ geothermal laboratory, Klaus Bauer for constructive discussions on data processing. We also thank all our colleagues from GFZ German Research Centre for Geosciences for their contribution to this work. The authors also acknowledge James Bailey, Mary Humphries and Colin Humphries for providing VSProwess software and sharing their extensive VSP data processing knowledge. Additionally, the authors thank the editor Dr Guilda Currenti and three anonymous reviewers for their comments and suggestions that helped to improve the paper.

425

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
