# Peer review of "3D deep geothermal reservoir imaging with wireline distributed acoustic sensing in two boreholes."

_Solid Earth, 2021_

## Referee Comment (RC2)

[referee-annotated manuscript omitted]

---

## Author Comment (AC1)

We would like to thank the anonymous reviewer who examined the resubmitted version of the manuscript for his/her additional comments and suggestions. We include the original comments followed by our reply.

In addition to changes based on reviewer's 1 and 2 comments, we performed an extensive work on discussion section and refined geological findings after discussion with colleagues who also worked on Rissdom-A project. Therefore, three additional co-authors who helped with data interpretation were included as co-authors Manfred Stiller, Ben Norden and Jan Henninges. The list of this additional changes can be found after the reply to the reviewer's comments.

**Remarks**

The references to pages and document lines are made using the document "clean" manuscript version.

**Anonymous Referee #1**

1. **The authors present results of a VSP experiment for geothermal explorations using DAS. This is an interesting topic and of great societal interest. The paper is generally well written and figures are broadly all relevant and of good quality.**

   **Before final publications I recoment that the authors clarify the overall purpose of the paper: Is it a DAS paper, a processing paper, or interpretation paper. I feel it is more of the latter. The overall question(s) that are adressed should be stated more explicitly in the introduction. Notably why and how VSP can help to make geothermal energy production in that particular area a success. Also, what are recommendations for future sites, both in similar and different geological settings. This would IMHO increase the "citability" of the manuscript.**

   Thank you for your comment. We clarified the objectives of the paper in the introduction section. In addition, in the discussion section we gave more explicit answers on the question regarding the value of the imaging results acquired using 3D DAS VSP at Groß Schönebeck.

**Minor comments:**

2. **Line 16: Referenced Spica 2020b before Spica 2020a**

   Fixed. (Page 2, lines 28-29).

3. **Line 25: The well is probably "completed", not just "drilled" ?**

   Words "**and cased**" added to the sentence. (Page 2, line 35).

   *Nevertheless, in a vast number drilled **and cased** boreholes, only measurements with a wireline logging cable are feasible.*

4. **L27: "...rarely found in [peer-reviewed] literature"...**

   "**peer-reviewed**" added to the sentence. (Page 2, line 37).

5. **L30: Add comment on how your work will contribute to geothermal exploration. What questions for decission makers are you adressing? What scientific quesitons are you adressing?**

   The objective of the paper was refined and clarified in the introduction section. (Pages 1-3, in particular lines 15-25, and lines 53-59).

6. **L63: "optimal" SNR. Is that optimum subjectively chosen? is there a quantitative approach to find the optimum? Is this published earlier and you could reference a figure?**

Gauge length is one of the essential parameters of the DAS data acquisition and processing. It influences the signal-to-noise ratio of the data and the resolution of the data. We applied an optimisation process as described by Dean et al. (2017). As a result, this approach means to help maximizing the signal-to-noise ratio and preserving the signal's frequency content, e.g. resolution of the data. The process of GL parameter optimisation was described in a previous publication Henninges et al. (2021). The reference to this publication is added to the text (Page 4, lines 97-98).

**7. L81: Please comment on if there are any DAS-specific steps neccesary (beside the polarity flip). strain(rate) data may be quite different from usual velocity data. Explicitly stating that no additional steps are required may help to promote this technology.**

Although wireline DAS technology offers a convenient and reliable method for seismic data acquisition, it has certain limitations as any geophysical method. The output physical quantity for the data acquired using DAS is strain or strain rate, not velocity as with conventional sensors like geophones. In case of data acquired at Groß Schönebeck, the recorded physical quantity was strain. Therefore, to achieve zero phase wavelet after correlation process a conversion to strain rate via time differentiation was required. The physics behind strain and velocity data is different. For instance, one of the known noticeable features of the data is the difference in polarity for reflections data and this is why we applied a polarity flip to compensate for that in the data. In addition, unlike conventional 3C-geophones and hydrophones, fibre optical cables have directional sensitivity patterns (Hartog et al. 2017; Wu et al. 2017; Martin et al. 2021). In practice, this means that the response function of the fibre for P-waves is described by $\cos^2(\theta)$ of the angle of incidence and $\sin(2\theta)$ for S-waves (Wu et al. 2017). In the experiment at the Groß Schönebeck in-situ laboratory in both boreholes, we acquired the data using straight fibres. To restrain the DAS method limitations related to a dependence on the angle of incidence during the experiment planning stage, a sophisticated ray tracing was performed to optimise VP positions and, as a result, the data quality for deep geothermal reservoir imaging. Thus, the influence of this method's limitation was significantly restricted. However, most shallow 500 meters for larger offsets have an absent P wave arrival. This happened because an angle of incidence for a seismic wave exceeds the critical angle for these shallow depths.

Another DAS feature is related to the nature of the recorded measurement. Stain or strain rate values are averaged over the interval called gauge length, discussed in a previous discussion point. Although optimisation suggested by Dean et al. 2017 theoretically should provide a balanced data in terms of signal-to-noise ratio and resolution, in specific cases smaller gauge length values are necessary to achieve imaging objectives. Larger gauge length values will cause smearing of the amplitudes over larger intervals, smaller values will preserve higher frequencies, but will have lower signal-to-noise ratio.

All the above-mentioned parameters should be carefully considered before applying data processing. In the case Groß Schönebeck data we had significant issues with amplitudes: ringing noise in the data, in addition variations caused by unknown reasons. Therefore, so far 3D DAS VSP dataset was only used for structural interpretation.

To use DAS VSP data for quantitative interpretation is not trivial because a good understanding and certainty in amplitude values is required. For instance, for calculation of FWI true amplitudes are required and DAS data even with conversion to velocity can not sufficiently provide it. Therefore, the results of such an inversion could be misleading. Thus, saying that only polarity flip is required to make DAS dataset directly comparable with the geophone data, is not correct.

**8. Table1: What do you mean by data conditioning?**

Data conditioning or preconditioning is a quite common wording and more or less a synonym for data preparation or pre-processing. It describes the application of some necessary processes to be carried out before the actual data-dependent processing can start. Such processes are e.g. geometry assignment, vertical stacking and correlation, non-subsurface related amplitude corrections etc, before essential processes like noise suppression, velocity analyses, CDP-stacking, migration etc. are practicable.

Since chapter 3.1 contains description of the ringing noise suppression, which is certainly more than simple conditioning, the subchapter was renamed into data conditioning, which is less specific than pre-conditioning or pre-processing (Page 6).

9. **L100: add references for Burg convolution and TF-attenuation**

Corresponding references were added to the text. (Page 6, lines 134-135).

10. **L104: "slapping" of the cable. Do you mean that loose bits of the cable are dangling around in the tubing and cause these signals. IT might need some additional explanation here.**

A more descriptive explanation and a reference to the paper on the ringing noise problem is added to the text (Page 6, lines 121-124).

11. **L119: "iteratively optimised" was that a manual or automatic process?**

Automatic process performed in the used software VSProwess X.

12. **L123: Was the ray tracing done in the anisotropic velocity model? Is it worth mentioning a reference to that ray tracer? is it publicly available**

A small anisotropic drift was included to the model. Commercial ray tracer included in the software from VSProwess Ltd. was used for computations. The reference to the software is included to the text (Page 8, lines 154).

13. **L134: "cleanest": subjectively or objectively?**

Visually cleanest result.

14. **Figure 4: The legend and text is too small**

The legend size and font for the text were increased (Page 10, Figure 4).

15. **Figure 5: 3D figures are difficult to understand in paper form. is that really neccesary here? would simple 2D slices be better?**

Due to the complexity of the case study, we think that 3D figures are beneficial for comprehension of the horizons distributions (for instance interlayering of the thin upper Rotliegend horizons) and visualisation of the differences between 3D surface seismic and 3D DAS VSP volumes.

16. **L191: the "Green arrow" is very hard to identify in the figure 6 (as are the other markers.**

The layout of the figure was changed. The size of the arrows was increased (Page 12, Figure 6).

17. **L205: Why is that sandstone formation chosen as target formation?**

Sandstone formation was chosen as target formation because of the good reservoir properties, porosity 8-10%, permeability 10-100 mD. (Page 15, lines 256-258).

18. **What do you expect to find with VSP that wasn't known from seismics? did you find this?**

    We expected from the log readings and core cuttings a part of Elbe sandstone reservoir section with improved properties. With DAS-VSP we were able to map thin Rotliegend horizons, internal structure inside the Elbe sandstone layer, lower Rotliegend unconformity. These features were previously not identified with 3D surface seismics.

19. **I suspect faults are relevant for reservoir integrity. What is known about faults in the area and can you in this paper help to comment on risks for geothermal production? I suspect the plan is for hydraulic stimulation?**

    One of the objectives of the 3D surface seismics and 3D DAS-VSP were to map possible discontinuities and fractures, which had previously been assumed to exist, based on interpretation of available legacy 2D seismic lines. Both cubes do not show direct indication of faults with displacement. The geological study of the area was conducted to ensure safe planning of the new well location and determine future stimulation targets (either sandstone layer or volcanites).

20. **Any risk for fault reactivation and major EQs?**

    Interesting question. This is beyond the scope of this paper, which is focused on subsurface imaging using active seismics (3D DAS VSP method).

21. **L271: Contractions shouldn't (sorry, should not) be used in scientific writing**

    Corrected. Don't changed to do not. (Sentence was moved to Page 4, lines 101-102).

22. **L324: it would be beneficial to posiiton your work in the broader geothermal exploraiton picture. Do you recoomend VSPs for all sites? What can be gained? What geological setting would justify it? Was DAS an adequate tool? Any lessons learned (in design, acquisition parameters, or processing)**

    An explanation of the broader context of the acquired knowledge from our case study was added in section 5.3 (Pages 19-20, lines 361-368). Lessons learned related to data acquisition/survey design) are described in section 5.1 (Page 18; lines 300-307, lines 317-321).

    **List of the additional changes to the manuscript:**

    1. Figure 3 (page 9). Bun**d**sandstein was corrected to Bun**t**sandstein
    2. Figure 3 (page 9. Z**ei**chstein was corrected to Z**e**chstein.
    3. Figure 4 (page 10). The position of the inline 104010 in Fig (a) was corrected slightly moved to SE
    4. Figure 8 (page 14). Position of the horizon R3 for XL 23825 was corrected.
    5. Throughout the text the name Elbe basis sandstone was changed to Elbe reservoir sandstone (ERS).
    6. Former bottom Elbe basis horizon interpreted from the DAS VSP data renamed into intra-bottom Elbe reservoir sandstone horizon (intra-bottom EBS).
    7. Throughout the text the name of the horizon R1 changed from top of Hannover Fm. to within Mellin-Schlichten R1.

8. Throughout the text the name of the horizon R3 changed from top Dethlingen Fm of to within Dethlingen Fm R3.
9. Definitions of the interpreted horizons introduced to the text.
   R1 (Page 14, lines 245-248).
   R3 (Page 14, lines 249-251).
   Top Elbe sandstone reservoir (Page 15, lines 280-282).
   Intra-bottom Elbe sandstone reservoir (Page 15, lines 269-274).
10. Acronym Slb changed to Schlumberger (Page 3, lines 85-86).
11. fine-to-coarse grained sandstone added on page 15, line 257 for the characterisation of Dethlingen Fm.
12. "This layer was deposited in aeolian setting and then reworked by aquatic processes." – was added for the characterisation of Dethlingen Fm. (Page 15, lines 258-259).
13. Contributions of additional co-authors has been added in author contributions section
14. Acknowledgement section was corrected, excluding co-authors of the current version of the manuscript.
15. Spelling of "signal-to-noise" ratio was made consistent throughout the text
16. Spelling of "upgoing" ratio was made consistent throughout the text
17. Spelling of "ray tracing" ratio was made consistent throughout the text
18. Small text corrections, such as changing from seismic to seicmics, articles corrections, typos corrections, language improvements (changes marked by red in case of deletion, blue in case of added words in revised version of the manuscript)

[revised manuscript text omitted]

---

## Author Comment (AC2)

We would like to thank the anonymous reviewer who examined the resubmitted version of the manuscript for his/her additional comments and suggestions. We include the original comments followed by our reply.

In addition to changes based on reviewer's 1 and 2 comments, we performed an extensive work on discussion section and refined geological findings after discussion with colleagues who also worked on Rissdom-A project. Therefore, three additional co-authors who helped with data interpretation were included as co-authors Manfred Stiller, Ben Norden and Jan Henninges. The list of this additional changes can be found after the reply to the reviewer's comments.

**Remarks**

The references to pages and document lines are made using the document "clean" manuscript version

**Anonymous Referee #2**

1. **Thank you for your submission. The manuscript shows an important application of seismic imaging at a geothermal site using distributed acoustic sensing (DAS). The manuscript shows a migrated 3D volume using DAS vertical seismic profiling and provides structural and stratigraphic interpretation of the main reflectors. This is one of the few applications using 3D VSP with DAS for characterization of geothermal sites.**
   **Overall, the manuscripts shows the necessary figures for its claims. However, I would suggest also adding a synthetic seismogram using well logs if possible. Additionally, I believe the paper needs careful proof reading to improve the readability and clarity. More especifically, the introduction should contain clear objectives, the claim and main manuscript outcomes. More work is needed in the literature overview to describe the previous research using DAS in geothermal sites and their achievements. Furthermore, I have added specific comments in the attached pdf.**

   Thank you for your comment. An additional figure with synthetic seismogram, well logs and seismic cross section through the boreholes was added to the manuscript. Following the recommendation, readability and clarity of the paper was improved. Introduction section was rewritten following the structure suggested in the supplementary comments document.

**Supplementary comments**

2. **I suggest re-working the Introduction as follows:**

   **(1) review - what is the problem? what has been done before to solve the problem? List the citations you have. This part needs more information regarding the use of DAS in geothermal fields.**
   **(2) claim - how is your work different from the previous work? How is it better and what are the challenges? Mention the limited use of DAS VSP with wireline fiber and how it can benefit characterization of geothermal fields. Describe your goals.**
   **(3) agenda - summarize what you will show in the paper and how it relates to your claim. Summarize your findings.**

   Introduction was rewritten following suggested structure (Pages 1-3).

3. **The authors provide a series of citations and applications, but more information on the applications of DAS in geothermal projects is needed. In particular, it's missing what has been done and what was found in the literature when it comes to DAS applications in geothermal fields. In addition to that, the authors should present why your application of DAS is different compared to what is seen in other geothermal fields and how it is helpful.**

Literature overview is added to the introduction section (Page 2, lines 38-52). Information on novelty of the presented work can be found on page 2; lines 53-58.

4. **what type of additional challenges?**

   More explanation added on page 2 lines 38-41.

5. **acronym should be inside parenthesis**

   Corrected (Page 4, line 90).

6. **Increasing the gauge length will have an effect similar to a spatial filtering, smoothing the signal. Can you add more discussing around how choosing 40 m gauge length might influence the resolution?**

   More discussion on the influence of the 40 m gauge length to the resolution is added on page 4, lines 94-98.

7. **Was the 3D VSP from both wells merged to create one volume?**

   Yes, correct. Data from both wells were processed uniformly and used as an input for the migration. For additional clarity name of the table changed to: 3D DAS VSP data processing flow for wells E GrSk 3/90 and Gt GrSk 4/05 (Page 5).

8. **It's not very clear to me what this step is? is this something like spectral whitening?**

   No, we used a moderate wavefield sharpening by a tau-p method (slant stacking). For clarity, in table 1 (Page 5) step "Data enhancement" was renamed to "Coherency enhancement", description was changed from "Moderate wavefield enhancement" to "Moderate wavefield sharpening by tau-p method".

9. **Too many paragraphs, which ended making for a difficult read. It could be condensed into one.**

   Formatting of the text was improved: paragraphs were merged into one (Page 6; lines 118-124).

10. **the fold map seems to be quite uneven, were there issues in acquisition or accessibility issues during the survey?**

    The original acquisition campaign was planned in 2 boreholes. Unfortunately, due to the cable failure event described (section 2.1, page 5 lines 106-109) the resulting coverage was significantly restricted. In addition, sentence on budget limitations shifted from the discussion section (Page 4; lines 101-102).

11. **The result after MPD reduces the relative amplitude quite significantly of what looks like is signal in comparison with the strong 3.7km reflection. This makes me think there could be some signal leaking after MPD? Or is there another explanation? For example, the strong reflection at 4.1km, present on both volumes, has much lower amplitude on the MPD result.**

    Substantial tests were performed to guarantee little to no amplitude leaking after the MPD. In Martuganova et al. 2021 it was demonstrated that suggested denoising successfully identifies resonances in the data and significantly improves the S/N without considerable influence on the signal.

    The observed differences in amplitudes were present on a first version of Figure 5 due to a few amplitude normalisation processes. 1) Due to uneven distribution of the bin density, data is normalised (divided by) bin density for migration to ensure more even amplitudes. 2) Amplitudes are normalised using RMS amplitude calculated in a window from 3400 to 4400 m.

In the updated version of Figure 5 (Page 10) we tried to improve amplitude balancing for the noisy cube such that the signals amplitudes are more equivalent and remaining differences are most of all MPD-caused.

**12. It would be helpful to point where the noise is.**

Horizon discontinuities pointed out by arrows; Noisy region highlighted by blue ellipse (Figure 5; page 10). Corresponding references to these pointers are added to the text (Page 11; lines 200-201).

**13. where is this shown?**

Size estimations added to Figure 4. In addition, text was modified accordingly (Page 10, Figure 4). In addition, the corresponding text modification was introduced on page 11, lines 203-205.

**14. It's difficult to follow this just from Figure 6.**

Additional ticks are added to the Figure 6 (Page 12), so the maximum depth of 4500 m is easier visible.

**15. are the vibration points for the surface seismic the same? or denser?**

Vibrator points locations are not the same. 3D surface seismic survey used a regular grid with source positions. The corresponding acquisition scheme can be found in Krawczyk et al. 2019 or Stiller et al. 2018.

**16. The fact that the binning of the surface seismic is 2x less dense seems to have been a choice in the processing, correct? The surface seismic doesn't have the same issues with illumination, I assume due to having denser shot points.**

The binning of the surface seismics is based on its acquisition geometry, i.e. source and receiver spacing. For the borehole data different bin size was chosen to achieve a compromise between sufficient bin fold and dense spatial sampling.

**17. please spell out your calculation.**

Description of the performed calculation was added to the text (Page 11, lines 211-217).

**18. are you able to produce a synthetic seismogram using sonic and density logs? how do these reflections compare with a synthetic seismogram?**

Additional figure (Figure 7, page 13) with synthetics, logs and 2D cross-sections intersecting wells was added to the paper.

**19. Please add 2D cross-sections intersection the wells, including the well tops. It is difficult to see from the 3D view where the top and base on the seismic.**

Additional figure (Figure 7, page 13) with synthetics, logs and 2Dcross-sections intersecting wells was added to the paper.

**20. On the 3D VSP, it looks like the reflection is at the top rather. Could you comment on this difference?**

This is the artifact of the 3D visualisation. Since the 3D cube is tilted and the inline is located on some distance from the borehole trajectory it creates such visual effect.

21. **I'm not sure if I fully understand this. When I look at Fig.4a, the blue rectangle includes areas of what seems to be zero coverage. Was any extrapolation used?**

   After discussions with co-authors originally suggested blue rectangle was changed to a complex polygon which follows the shape of the data better (See Figure 4a; page 10). This allowed to clip out excessive areas of extrapolation. For gridding convergent interpolation method (Petrel software) was used with a grid cell size 12.5 to 12.5 m.

22. **please make sure you check the journal's rules for citing manuscripts in preparation.**
   Reference (Norden et al., in preparation) was changed to pers.comm. and removed from the reference list.

23. **for this project or overall?**

   For the project. "within this project" was added to the sentence for clarification purposes (Page 18, line 303).

24. **This statement is a bit unclear. Based on what?**

   "with permanently installed fibre optic cable" added to the sentence for clarification (Page 18, line 315).

25. **I don't fully agree with the low cost because the wireline cables can be quite expensive and you still count on rig time. It could be possibly lower the cost when compared to geophone VSP.**

   "in comparison with VSP acquired using geophones" added to the text (Page 18, line 318).

26. **could you comment on the coupling mechanisms of the wireline fiber?**

   Details on survey, including cable installation can be found in the section 2.1 In addition, "and hanging freely with 1 m slack applied" added to the text for additional clarity (Page 3, line 87-88).

27. **I agree the authors present a unique dataset in the field of geothermal, specially when it comes to 3D VSP acquired on wireline data, which is a very unique niche itself. However, there have been other examples of seismic data acquired with DAS at geothermal fields. One example is the FORGE site, in Utah, USA, to detect microseismic events and to record guided waves. I suggest the authors add a more comprehensive literature overview in particular to geothermal applications.**

   Literature overview was added in the introduction section (Pages 1-2).

28. **specify what are good results.**

   Good results change to "A detailed imaging of the target reservoir interval (with vertical resolution up to 16 m)" (Page 18-19, lines 332-333).

29. **I suggest adding this information to Section 2.1 for clarification on the data acquisition.**

   Sentence related to budget limitations was moved to section 2.1 (Page 4, lines 101-102).

30. **This belongs to the literature overview.**

   This section was modified to shift more into the direction of the comparison of our survey with other known DAS VSP campaigns (Page 19, lines 338-347).

**31. This is new information in the discussion - some evidence or citation to previous work must be added.**

This is a conclusion for our dataset specially. "for our dataset" was added to the text. I hope that this clarifies the issue (Page 19, lines 357).

**32. move to the beginning of section**

The paragraph was rewritten according to latest discussions on geological data (Page 20, lines 377-382).

**33. can you comment on possible sub-seismic faults?**

Thank you for a good question. Sub-seismic faults are important for the geothermal energy reservoirs, since they are often considered as a pathways for fluid flow. One of the scientific objections for this detailed imaging using vertical seismic profiling was to study local discontinuities elements in the reservoir section. As a result, neither borehole seismic cube nor surface seismic cube detected apparent faults or fractures. Nevertheless, although the DAS seismic cube had a higher resolution than same cube from 3D surface seismic, it is still limited at these large depths.

First, the created fracture by the hydrofracturing cannot be imaged unless there is a distinct acoustic contrast or offset detectable by seismic methods. For 3D DAS VSP imaging, gauge lengths of 20 and 40 m were used. These values might be too high to detect small scale features, such as fractures. Theoretically, a smaller gauge length is necessary for small scale details imaging, such as fractures, but it comes at the expense of having a lower signal-to-noise ratio, which is important at these large depths.

To achieve a sharper and more detailed image at the Groß Schönebeck, it is necessary to further improve the velocity model used for migration. Sophisticated anisotropy study and inclusion of the heterogeneities into the existing layered velocity model will allow to further improve the reservoir image. This might be especially important for the potential fracture mapping in the case of data acquisition with active seismic methods.

To better understand the expected response from the fractures, it is essential to perform modelling to predict the wave propagation in the fractured reservoir. It could help to estimate the size, contrast required to image a fracture, characterise, and interpret the individual response with such modelling. Further modelling studies are needed to evaluate necessary acquisition parameters.

**34. I suggest removing this or adding an explanation why you consider still a new method.**

"is still a relatively new method" was removed from the text (Page 21, line 407).

**List of the additional changes to the manuscript:**

1. Figure 3 (page 9). Bun**d**sandstein was corrected to Bun**t**sandstein
2. Figure 3 (page 9. Z**ei**chstein was corrected to Z**e**chstein.
3. Figure 4 (page 10). The position of the inline 104010 in Fig (a) was corrected slightly moved to SE
4. Figure 8 (page 14). Position of the horizon R3 for XL 23825 was corrected.
5. Throughout the text the name Elbe basis sandstone was changed to Elbe reservoir sandstone (ERS).
6. Former bottom Elbe basis horizon interpreted from the DAS VSP data renamed into intra-bottom Elbe reservoir sandstone horizon (intra-bottom EBS).
7. Throughout the text the name of the horizon R1 changed from top of Hannover Fm. to within Mellin-Schlichten R1.

8. Throughout the text the name of the horizon R3 changed from top Dethlingen Fm of to within Dethlingen Fm R3.
9. Definitions of the interpreted horizons introduced to the text.
   R1 (Page 14, lines 245-248).
   R3 (Page 14, lines 249-251).
   Top Elbe sandstone reservoir (Page 15, lines 280-281).
   Intra-bottom Elbe sandstone reservoir (Page 15, lines 269-274).
10. Acronym Slb changed to Schlumberger (Page 3, lines 85-86).
11. fine-to-coarse grained sandstone added on page 15, line 257 for the characterisation of Dethlingen Fm.
12. "This layer was deposited in aeolian setting and then reworked by aquatic processes." – was added for the characterisation of Dethlingen Fm. (Page 15, lines 258-259).
13. Contributions of additional co-authors has been added in author contributions section
14. Acknowledgement section was corrected, excluding co-authors of the current version of the manuscript.
15. Spelling of "signal-to-noise" ratio was made consistent throughout the text
16. Spelling of "upgoing" ratio was made consistent throughout the text
17. Spelling of "ray tracing" ratio was made consistent throughout the text
18. Small text corrections, such as changing from seismic to seicmics, articles corrections, typos corrections, language improvements (changes marked by red in case of deletion, blue in case of added words in revised version of the manuscript)

[revised manuscript text omitted]

---

## Author Response (AR2)

We would like to acknowledge the work of the topical editor Guilda Currenti and the anonymous reviewer for their comments and suggestions that helped to improve the paper. We include the original comments followed by our reply.

The references to pages and document lines are made using a "clean version" of the manuscript.

**Editor's note**

Dear Author,

I am glad to announce that the manuscript could be accepted to final publication after minor revisions. Please, address the points raised by the second referee in his report and my suggestions reported below.

Best Regards

Gilda Currenti

**Editor Comments**

1. **Figure 8 panel b: The inlines are reported as IN. Please, correct "IL 103995" in "IN 103995", if it is the inline shown in the inset. Otherwise explain what IL is.**

   IL 103995 was corrected to IN 103995. (Page 14, Figure 8)

2. **Figure 9 panel b: As in Fig 8, please correct "IL 103975", if it is appropriate.**

   IL 103995 was corrected to IN 103995. (Page 16, Figure 9)

3. **I suggest to update the reference list about the use of DAS in volcanology, adding the recent published papers:**

   The list of publications for DAS applications in volcanology was updated according to the suggestions listed below. (page 2, line 27)

   **Klaasen, S., Paitz, P., Lindner, N., Dettmer, J., & Fichtner, A. (2021). Distributed acoustic sensing in volcano-glacial environments—Mount Meager, British Columbia. Journal of Geophysical Research: Solid Earth, 126, e2021JB022358. https://doi. org/10.1029/2021JB022358**

   **Jousset, P., Currenti, G., Schwarz, B. et al. Fibre optic distributed acoustic sensing of volcanic events. Nat Commun 13, 1753 (2022). https://doi.org/10.1038/s41467-022-29184-w**

   **Currenti, G., Jousset, P., Napoli, R., Krawczyk, C., and Weber, M.: On the comparison of strain measurements from fibre optics with a dense seismometer array at Etna volcano (Italy), Solid Earth, 12, 993–1003, https://doi.org/10.5194/se-12-993-2021, 2021.**

**Report #2 from Anonymous referee #3**

**Dear Authors,**

**Thank you for submitting your interesting work to Solid Earth. Indeed, applying novel methods, such as DAS, to carbon-neutral energy projects, such as geothermal, are of great interest and importance.**

**The paper demonstrates the successful application of 3D DAS VSP imaging using wireline fibres. In addition, the authors did great work addressing the issues and challenges related to such dataset and showed the value of the received image for a decision on further wells placement for harvesting geothermal energy.**

**The paper is well written, the abstract is to the point, and the conclusion provides a complete summary of the presented work. Especially I want to praise the authors for their work on the quality of illustrations and detailed explanation of the processing steps in Table 1.**

**Below is the minor suggestions which may further improve this work:**

1. **24-25: "measure particle displacement" -- despite the usage of the hDVS system, it is commonly assumed that DAS measures strain change or strain rate. Please clarify.**

   "measure particle displacement" was changed to "measure strain". Page 2, line 24.

2. **47: it would be beneficial to mention reservoir temperature in the considered case to understand if high-temperature geophones could be used or not.**

   "in the subsurface with elevated temperatures up to 175°C" was added to the text on page 2, line 43.

3. **111: it would be great if authors could bring their hypothesis on "unknown reasons"**

   We added some explanation regarding what possibly caused amplitude heterogeneities in the data. These two sentences were added to the text: "This behaviour could be related to the local repositioning of the cable inside the borehole since similar reduced-amplitude patterns were observed in the recordings with extra slack provided to the cable (c.f., Henninges et al., 2021). Further research is required for a systematic understanding of the here qualitatively explained effects.." (Page 5, lines 110-113)

4. **Figure 2: it could be beneficial to highlight the changes in the seismograms, which the reader should pay attention to between processing steps.**

   We added colour-coded arrows to point to intervals affected by the ringing noise, downgoing P-wave arrival and reflections (Page 7, Figure 2). In addition, corresponding references were added to the text, describing Figure 2: Page 5, lines 130 and 132, page 6, lines 135 and 145-146. Also, we better refined the already mentioned noisy intervals and added a few new ones: page 5, line 131 and page 6, line 135.

5. **157: were Thomsen's parameters chosen only based on DAS VSP data, or some logs were used to estimate them? Please elaborate.**

   Thomsen's parameters were chosen only based on DAS VSP data. This reduced the standard deviation of drift for many of the longer offset VPs. It should be noted that this was not a comprehensive anisotropy study, and the parameters used are only a guide or fix to make better the data fit the layered model. (Page 8, lines 161-162)

6. **Figure 3, 4b, 5, 6: North arrow is confusing. Consider changing the north arrow on 3D plots or using northing and easting if x and y coordinates match these orientations. Also, some figures have "N" on top of the arrow; some do not.**

A more straightforward and more intuitively understandable North arrow sign was used for updated Figures 3 (Page 9), 4b (Page 10), 5 (Page 10) and 6 (Page 12).

7. **304: "constelation fibres' -> "engineered fibres", as constelation is a trademark**

Corrected. Page 18, line 307.

8. **325: "geothermal fiel, which are limitd to the depth up to 600 m." -> "geothermal field, which are limited to the depth of 600 m."**

Corrected. Page 18, line 329.

9. **399: 'borehole seismics" -> "borehole seismic"**

Seismic is an adjective. We believe that the noun "seismics" should be used in this sentence. Therefore, the sentence was not changed. Page 21, line 400.

**I recommend accepting the paper for publication after incorporating these minor suggestions.**

Thank you. We followed all suggestions for corrections as close as possible to clarify the residual issues with the manuscript.

Meanwhile, the paper on 3D surface seismic interpretation/ geological modelling was submitted and was published as a preprint in Geothermal Energy. Therefore, we change the reference B. Norden, personal communication, April 28, 2022, to Norden et al., in revision (Page 11 line 210, Page 13 line 238, page 14 line 252, page 15 line 266, page 17 line 288, page 20 line 376-377 and 386-387). The corresponding reference was added to the reference list:

Norden, B., Bauer, K., Krawczyk, C.M., 2022. From pilot site knowledge via integrated reservoir characterization to utilization perspectives of a deep geothermal reservoir: 3D geological model at the research platform Groß Schönebeck in the Northeast German Basin. Geothermal Energy, in revision, doi:10.21203/rs.3.rs-1660889/v1

In addition, we added two sentences to the acknowledgement section: "The publication of this work is supported within the funding programme "Open Access Publikationskosten" Deutsche Forschungsgemeinschaft (DFG, German Research Foundation) - Project Number 491075472." and "Additionally, the authors thank the editor Dr Guilda Currenti and three anonymous reviewers for their comments and suggestions that helped to improve the paper."